# Study on repair materials and technologies for addressing crack-related damage in the Earthen City Wall of Kaifeng

Tingting Yue[1]*, Xizhi Zhang[1,2], Jianwei Yue[3], Wenhao Li[3], Xiang Zhu[3], Jingwen Yue[4]

1 Tianjin University School of Civil Engineering, Tianjin University, Tianjin, China, 2 Architectural Design and Planning Research Institute Co., Ltd., Tianjin, China, 3 School of Civil Engineering and Architecture, Henan University, Kaifeng, China, 4 Zhengzhou University School of Archaeology and Cultural Heritage, Zhengzhou University, Zhengzhou, China

* 781349393@qq.com

## Abstract

Rain and snow seepage into the cracks of the soil wall is the leading cause of its surface weathering, and the key to crack repair lies in developing reasonable repair materials. Based on the carbonation principle and mineralization mechanism of quick-lime, this study focuses on the cracks in the city wall of Kaifeng as the research subject. Urea urease solution, quicklime, sodium methylsilicate, styrene-acrylic emulsion, waterborne polyurethane, and soil were selected to prepare 27 groups of high-fluidity repair materials with varying proportions. The surface strength, water absorption, surface strength after water absorption, consistency and freeze-thaw cycle tests of crack repair samples were carried out to explore the repair effects of different proportions of repair materials on crack diseases. The results demonstrated that the urease urea solution, along with its mineralization reaction with quicklime and sodium methylsilicate, significantly accelerated the chemical interaction between quicklime and sodium methylsilicate, thereby enhancing the mechanical and waterproofing properties of the repair materials. The group composed of 7.5% quicklime and 7% sodium methylsilicate exhibited a total efficacy coefficient of 96.25, indicating superior mechanical strength and waterproofing performance. Among the tested waterproofing agents—sodium methylsilicate, styrene-acrylic emulsion, and waterborne polyurethane—the effectiveness was ranked as follows: waterborne polyurethane > sodium methylsilicate > styrene-acrylic emulsion. Notably, the group containing 5% sodium methylsilicate combined with 7.5% quicklime achieved the lowest water absorption rate of 3.9%. Thirty cycles of crack-filling experiments revealed that the crack resistance of the filling material surpassed that of the original sample, while maintaining excellent integrity with the earthen structure even after freeze-thaw cycles (Fig 1).

**Data availability statement:** All relevant data are within the manuscript.

**Funding:** The author(s) received no specific funding for this work.

**Competing interests:** The authors have declared that no competing interests exist.

## 1. Introduction

As a crucial component of ancient Chinese architecture, the earthen city wall embodies rich historical, cultural, and engineering significance. It stands as one of the most important representatives of immovable cultural heritage in China. However, the majority of existing earthen walls are primarily composed of silty clay, and their construction techniques predominantly involve direct compaction of raw soil or raw adobe. When exposed to outdoor conditions, these walls are frequently subjected to damage from rainfall and flooding, leading to surface crusting [1]. Subsequently, cracks emerge, which significantly impact the stability and durability of the heritage sites. According to Shen Wenwu's research, the melting of snowfall exerts a significant impact on the strength of earthen city walls, with the most pronounced damage occurring at the top. Additionally, as water infiltrates the soluble salts within the earthen structure, it induces further deterioration of the wall [2]. Richards employed a portable wind-rain erosion simulation device to investigate the comparative impacts of wind, sand-laden wind, and wind-rain on the degradation of soil-based heritage. Furthermore, he elucidated the correlation between the early degradation characteristics of test walls induced by these factors and the degradation patterns documented on historical city walls [3]. Yue Jianwei et al. employed an X-ray diffractometer and a scanning electron microscope to analyze the chemical composition and microstructure of soil samples from earthen sites. Using the strength reduction method, they determined the minimum safety factor of trenches under both natural conditions and following rainfall exposure. It was found that rainfall erosion, freeze-thaw cycle, and artificial destruction were the main factors leading to the degradation of earthen sites [4]. As the water content increases, the safety factor of earthen ruins decreases, and water exerts a significant influence on the mechanical strength of these structures. Soil exhibits large pores, numerous initial cracks, and a tendency toward easy spalling. Richards et al. employed a newly developed cellular automata model to evaluate the risk of environmental degradation over 100 years period under a range of potential climate and protection scenarios. It has been found that an increase in wind speed significantly elevates the risk of overall deterioration, while the risk associated with rainfall-driven degradation characteristics is even more pronounced [5]. This paper investigates the impact of floods on earthen architectural heritage and proposes a qualitative methodology for assessing the flood vulnerability of such structures. Through a case study of the Iberian Peninsula, this paper analyzes the visible damage to soil structure buildings associated with superstructure-soil interaction, as well as the damage caused by flooding, to emphasize the fragility and resilient resources of this cultural heritage. Finally, a flood control strategy and protection standard for soil architectural heritage are proposed [6]. Cavicchio discovered that local damage, caused by cracks and material losses, rendered the original wall more vulnerable when exposed. Additionally, the intensity of precipitation events has increased as a result of climate change, thereby providing a foundation for practical research on the shelter coat of Union Fort. The optimal properties of repair materials encompass excellent consistency (plasticity), minimal shrinkage, strong adhesion to the substrate (durability), high cohesive strength against erosion (durability), low

liquid water absorption, high desorption capacity, moderate water vapor permeability, and equivalence in color and texture to the substrate [7]. It is evident that precipitation, such as rain and snow, infiltrates into the soil city wall, leading to a reduction in the strength of its surface layer. Under the combined effects of scouring, evaporation, freezing, and thawing, this results in pulverization and spalling of the wall Fig 1.

Due to its long geological age, low strength, and exposure to an outdoor natural environment, the earth wall is susceptible to surface degradation, particularly the further development of cracks, which has increasingly drawn the attention of researchers. Liu Shiyu and Liu Xiaojun utilized microbial-induced calcite precipitation (MICP) technology to repair cracks. They discovered that the integrity and mechanical properties of the samples were enhanced through the microbial remediation method [8,9]. Yuan Pengbo utilized enzyme-induced calcium carbonate precipitation (EICP) technology to reinforce the cracks in earthen sites in Northwest China. The study found that this restoration method can effectively reduce dynamic strain generation, increase soil density, and enhance structural strength [10]. Through research conducted by Cui Kai, it was found that the pile-slurry cooperative reinforcement of site cracks can effectively address the issue of dry shrinkage on the bonding surface and enhance the integrity of the structure after repair [11–13]. A SH-(CAO + C + F) remediation solution, composed of quicklime as an admixture, 1.5% SH binder, a specified amount of clay, and fly ash, is proposed. The remediation solution exhibits excellent physical and mechanical properties, as well as good compatibility with earthen ruins. Zhang Jingke utilized gravel and wall soil as the primary materials for repairing cracks in the city wall. Through experimental analysis, it was found that this slurry exhibits the characteristics of rapid strength enhancement and stable physical properties. Through the research conducted by numerous scholars, it is evident that most cracked earthen sites are repaired using high-fluidity materials. The integrity and durability of these sites are enhanced by consolidating the repair materials with surrounding soil or inducing calcium carbonate precipitation through microbial activity [14].

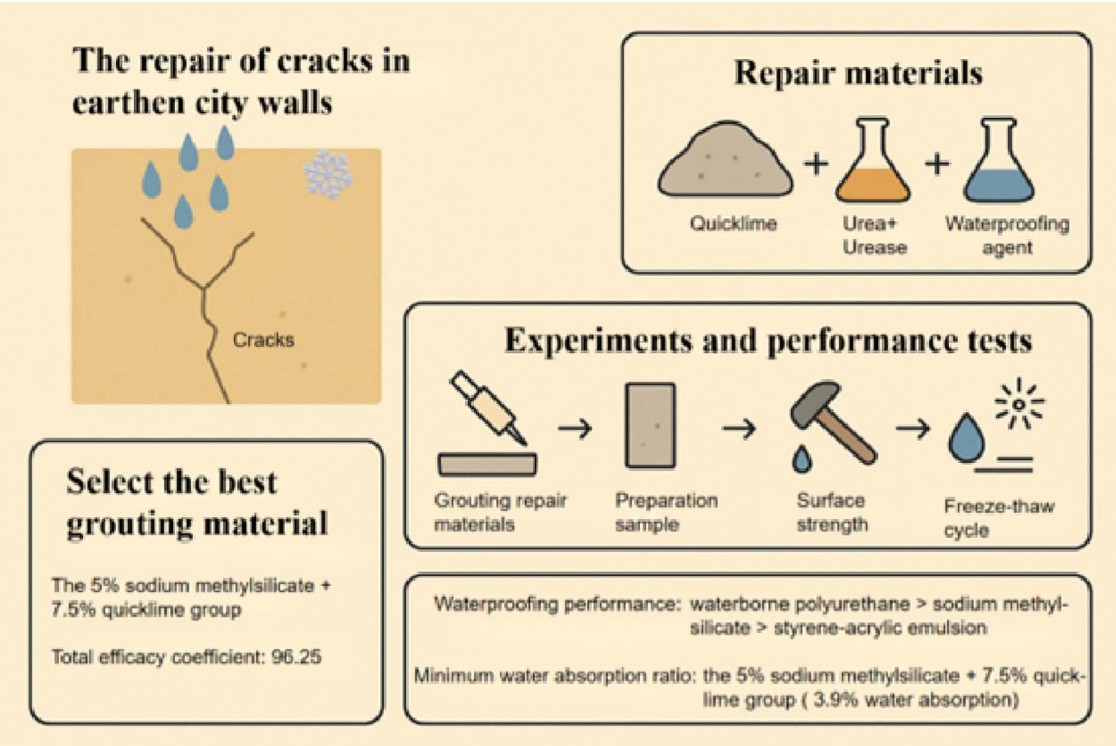

**Fig 1. Distribution Curve of Particle Size.**

In addition to enhancing the mechanical properties of earthen sites through the mineralization principle, scholars have also conducted research on the hydraulic properties of these sites to improve their waterproof performance. Yue Jianwei et al. incorporated a silicone waterproofing agent into the restoration materials to enhance their hydrophobicity, resulting in a significant improvement in hydrophobic performance [15]. Wang et al. developed a permeable polyurethane grouting material and incorporated water-soluble polyurethane as an admixture. The results demonstrated that polyurethane can significantly enhance both the bearing capacity and impermeability of soil, making it a highly effective admixture [16]. Xu developed a range of high-performance geopolymer grouting materials using slag and fly ash as raw materials and basalt fiber as an admixture. It was found that basalt fiber can enhance the workability of grouting materials while reducing the drying shrinkage of the materials [17,18]. However, basalt fiber is primarily utilized for the repair of existing structures, which fails to satisfy the restoration requirements of historical buildings. Li et al. investigated the performance of polymer mortars (including fiber-reinforced mortars and acrylic emulsion cement mortars) utilized for repairing ancient masonry structures damaged by the 2008 Wenchuan earthquake. They tested the compressive strength, tensile strength, carbonation resistance, and weathering performance of simulated polymer mortar specimens [19]. The results indicated that acrylic emulsion and epoxy resin mortars exhibited high mechanical properties and durability. Prakash B. et al. employed MICP technology to repair cracks in cement mortar with widths ranging from 0.12 to 1.3 mm. The crack widths effectively repaired were within the range of 0.29 to 1.1 mm, indicating that MICP is a highly effective repair method. However, since the experimental process requires soaking the sample in an MICP solution, this poses challenges for implementation in actual repair processes [20]. Ma et al. incorporated sodium methylsilicate and organosilicon into quicklime, thereby enhancing its durability and forming a hydrophobic layer on the surface of modified soil particles to further improve their erosion resistance and water resistance [21]. However, this paper does not address whether the hydrophobic layer formed by it would affect the air permeability of the sample itself and potentially lead to secondary failure. Mu et al. utilized MICP technology for the repair of historical city wall bricks, mixing the MICP solution with silica powder and directly injecting it into cracks [22]. The results demonstrate that MICP is capable of effectively repairing both natural cracks and simulated cracks on brick surfaces. Liu et al. carried out researched on superhydrophobic coatings for the surface of earthen heritage sites. They emphasized the potential of such coatings for terms of their waterproof and moisture-resistant properties. Nevertheless, these materials typically place more emphasis on the surface coverage effect, with relatively less attention given to crack filling repairs and the durability within the structure [23]. Li et al. improved the capillary water rise and water contact angle of silty soil by applying silicone-based waterproofing agents, achieving an inhibition rate of nearly 98%. This research verified the significance of adding a small quantity of silicates in enhancing the waterproof performance [24]. Cheng et al. conducted a systematic evaluation of polyurethane grouting materials. These materials exhibited excellent penetration ability and adhesiveness in crack repair and grouting filling. In particular, their adaptability on complex interfaces was improved. Although this study was aimed at the repair of concrete structures, the use of polymer waterproof agents and organic materials therein provides a reference for soil restoration [25].

Numerous scholars have delved into the reinforcement and conservation of earthen heritage sites. Their research outcomes have significantly propelled the advancement of restoration technologies and materials for such sites. However, the majority of current studies predominantly concentrate on the application of single restoration materials (such as quicklime, sodium methylsilicate, polyurethane, and styrene-acrylic emulsion) or individual techniques (such as MICP/ EICP). There is a notable scarcity of systematic evaluations of composite restoration materials. In particular, there is a lack of research on crack grouting that takes into account both hydraulic and mechanical properties concurrently. In response to this gap, this study has designed and tested 27 combinations of quicklime, soil, and admixtures, thereby furnishing a certain experimental foundation for the repair of cracks in earthen city walls.

## 2. Experimental materials

The objective of this study is to develop materials specifically designed for repairing cracks in earthen ruins. It is anticipated that the developed materials will possess characteristics such as excellent fluidity, water resistance, enhanced

crack resistance, superior strength, and a color that closely resembles that of earthen city walls. To achieve these properties, it is essential to compound multiple materials. Therefore, based on preliminary testing and a comprehensive literature review, along with the requirements for cultural relics preservation, we have selected two traditional materials commonly used in earthen city walls: soil and quicklime. Additionally, we identified three waterproofing agents-urea urease and sodium methylsilicate, waterborne polyurethane, and styrene-acrylic emulsion-to develop effective restoration materials.

## 2.1. Basic indicators of soil samples

### (1) Distribution of Particle Sizes

To analyze the particle characteristics and distribution of the soil utilized in this study, a particle analysis test was conducted. This test determined the proportions of particles across various size categories. The particle size of the soil samples utilized in the test ranges from 0.075 mm to 60 mm. After the drying treatment, soil samples with a particle size of less than 2 mm are selected for testing. In the test, a standard vibrating screen machine was employed for screening purposes. This machine operated at a frequency of 220 shakes per minute, while the actual number of vibrations recorded was 145 times per minute. When utilizing data, it is essential to first record the quality of the empty disk. Subsequently, subtract this recorded quality from the data and incorporate the result into the following formula:

$$X = \frac{A}{B} \times 100$$

(1)

In the formula, X represents a value that is less than the proportion of soil samples exhibiting a specific particle size relative to the total mass of the sample (%);

A-the total mass of soil particles less than a specific particle size (g);

B-total particle mass (g).

To enhance the clarity and intuitiveness of the test data, the data information is represented along the X and Y axes. A particle gradation diagram is then generated using Origin software, employing semi-logarithmic coordinates (Fig 2).

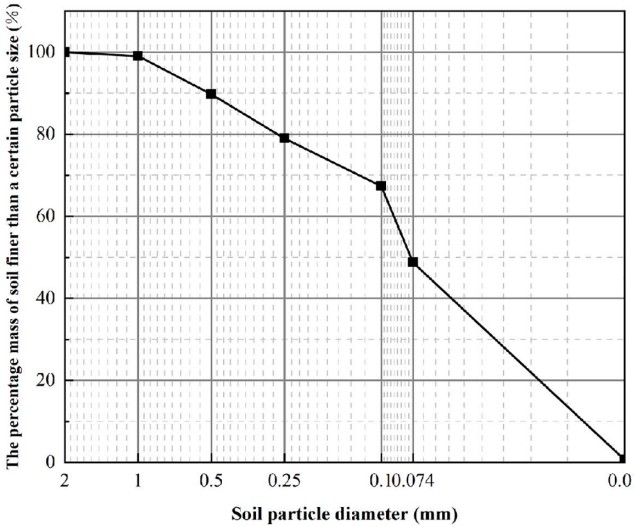

**Fig 2. Distribution curve of particle size.**

## (2) Refine water cut limits

To more accurately determine the liquid-plastic limit, liquid limit index, and plasticity index of the soil in question, relevant data were collected through a measurement test for limit water content. The plasticity index is determined by the difference between the liquid limit and the plastic limit. The soil utilized in this assessment is classified accordingly. The limit water content is determined using the liquid-plastic limit tester method. The specific procedure is carried out in strict accordance with the steps outlined in the relevant standard. The equipment utilized in the test is a digital display soil liquid-plastic limit tester, model LP-100D. The falling cone has been selected for marking purposes. When the falling cone reaches the designated depth, the moisture content of the sample is assessed to determine both the liquid limit and plastic limit of the soil utilized in this test. The advantage of employing this method lies in its ability to mitigate the impact of human factors, while ensuring that the entire operational process is straightforward and easy to master. The computed values are presented in Table 1.

## (3) Compaction test

The objective of this experiment is to determine the optimal moisture content and maximum dry density of soil samples. These two indices represent a set of relative parameters that are established within a specific compaction environment. The relationship between dry density and moisture content is characterized by a positive correlation, provided that the compaction method and conditions remain constant. When the dry density reaches its maximum value, a negative correlation emerges between the two variables, indicating a quadratic linear relationship. The moisture content at the apex represents the optimal moisture level, while the corresponding dry density is identified as the maximum dry density [26].

As illustrated in Fig 2 and Table 1, the soil utilized in this experiment is classified as silty clay, characterized by relatively fine particles. Therefore, in accordance with the testing requirements, a light compaction method was employed for the compaction tests of the soil samples. The diameter of the compaction barrel utilized in the test is 100 mm, in accordance with relevant operational standards. Electric Numerical Control Compaction Instrument JDS-1, manufactured by Nanjing Soil Instrument Factory. The detailed testing methodologies are outlined as follows:

1. Prepare five portions of test soil with a particle size of less than 5 mm, followed by drying and crushing;

2. According to the measured limit water content from the test, each group is increased by 2%. Five groups of samples with varying water content are prepared and sealed for a soaking period of 12 hours.

3. Compact each group of samples thoroughly. The process involves three distinct layers of compacted samples, with each layer undergoing compaction a total of 25 times. Following the compaction of one layer, it is essential to scrape off any excess material before re-filling with soil and subsequently repeating the compaction for the preceding layer.

4. After the compaction process is completed, the soil sample should be demoulded. Excess material at the upper end of the compaction cylinder must be carefully scraped off using appropriate tools, followed by measuring both the mass and density of the sample.

5. Obtain a sample from the center and dry it in an oven for 8 hours. Subsequently, calculate the moisture content using Equation 2:

**Table 1. Test Results of Limit Water Content.**

| Plastic limit $\omega_P$ (%) | Liquid limit $\omega_L$ (%) | Plasticity index $I_P$ | Refinement of Soil Classification |
|---|---|---|---|
| 23.1 | 38.2 | 15.1 | Silty clay |

$$\omega_0 = \left(\frac{m_0}{m_d} - 1\right) \times 100$$

<div align="right">(2)</div>

Where, $m_d$--dry soil weight (g) of the sample;

$m_0$--Wet soil weight (g) of the sample.

6. Refinement of the Dry Density Calculation for Soil Samples:

$$\rho_d = \frac{\rho_0}{1 + 0.01\omega_i}$$

<div align="right">(3)</div>

Where, $\rho_d$--dry density of sample (g/cm³);

$\rho_o$--Wet density of the sample (g/cm³);

$\omega_i$--Moisture content (%) of the sample at a certain point.

The measured data are presented in Table 2, and the relationship between moisture content and dry density of the soil utilized in the test is illustrated using Origin software, as depicted in Fig 3.

**(4) X-ray fluorescence (XRF) analysis of soil samples**

XRF spectrum analysis is an effective method for determining the elemental composition of samples. This technique offers several advantages, including non-destructive testing, the ability to analyze a wide range of elements, high accuracy in element detection, minimal risk of secondary contamination, and low detection costs. The principle of this method involves the interaction of X-rays with the substance being analyzed, resulting in specific characteristics of X-ray fluorescence intensity. By utilizing corresponding curves that illustrate the relationship between these characteristic intensities and the

**Table 2. Correlation between Moisture Content and Dry Density.**

| Moisture content % | 10 | 12 | 14 | 16 | 18 |
|---|---|---|---|---|---|
| Dry density g/cm³ | 1.65 | 1.69 | 1.72 | 1.76 | 1.72 |

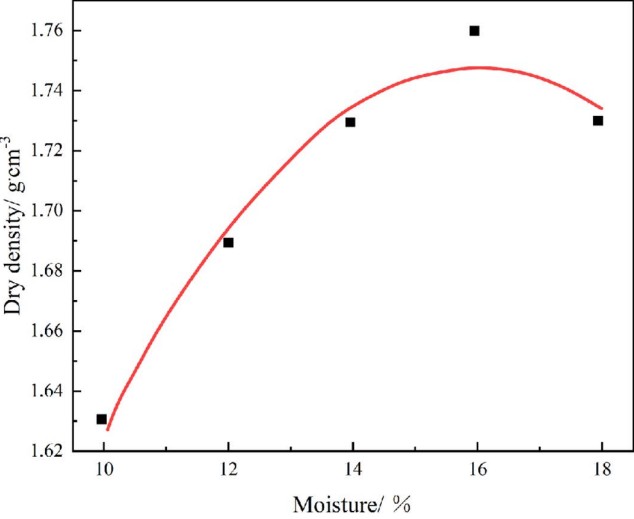

**Fig 3. Curve graph.**

concentrations of various elements, it is possible to calculate the different components present within those elements. This technique is employed to analyze the components of the deteriorating section of the Kaifeng city wall.

The instrument utilized for this analysis is the X-ray fluorescence spectrometer, manufactured by Thermo Fisher Scientific, a prominent company based in the United States. The results of the elemental measurements are presented in Table 3 and illustrated in Fig 4.

## 2.2. Restoration materials

### (1) Quicklime

Quicklime was extensively utilized in the construction of earthen structures during ancient times. It was frequently combined with glutinous rice paste, river sand, and soil to create a composite material known as triple soil. Consequently, quicklime has been chosen as one of the restoration materials discussed in this paper. The primary component of quicklime is calcium oxide (CaO), which reacts with water and carbon dioxide to produce calcium hydroxide. This compound can further undergo carbonation to form calcium carbonate, effectively filling the voids between soil particles and enhancing the cohesion among material particles. Due to the rapid pace of construction, the newly restored earthen site is likely to experience deterioration, including issues such as detachment and cracking.

Quicklime is manufactured by Kemio Chemical Reagent Co., Ltd. The physical and chemical specifications are as follows: the relative molecular weight is 56.08, and the CaO content is not less than 98%. After conducting thorough investigations and reviewing relevant literature, this paper proposes the use of quicklime to enhance the structural integrity of earthen ruins in the Kaifeng area. Quicklime is employed not only for the restoration of existing earthen ruins but also for newly constructed city walls. Consequently, quicklime serves as a key component in the formulation of restoration materials for earthen city walls discussed herein. When quicklime is combined with soil, it undergoes a reaction with the moisture present in the soil as well as with carbon dioxide from the atmosphere:

$$CaO + H_2O \rightarrow Ca(OH)_2 \tag{4}$$

$$Ca(OH)_2 + CO_2 \rightarrow CaCO_3 + H_2O \tag{5}$$

**Table 3. Composition of Materials and Elements in Test Soil.**

| compound | Content (%) | element | Content (%) | compound | Content (%) | element | Content (%) |
|---|---|---|---|---|---|---|---|
| $SiO_2$ | 57.80 | Si | 27.42 | $ZrO_2$ | 0.017 | Zr | 0.0140 |
| $Al_2O_3$ | 16.04 | Al | 8.09 | $Rb_2O$ | 0.0134 | Rb | 0.012 |
| CaO | 10.50 | Ca | 7.32 | $Cr_2O_3$ | 0.0160 | Cr | 0.0122 |
| $Fe_2O_3$ | 6.18 | Fe | 4.51 | NiO | 0.0064 | Ni | 0.0071 |
| MgO | 3.10 | Mg | 2.09 | CuO | 0.0055 | Cu | 0.004 |
| $K_2O$ | 1.6 | K | 1.03 | $As_2O_3$ | 0.006 | As | 0.0047 |
| $Na_2O$ | 1.47 | Na | 1.21 | I | 0.0043 | I | 0.0048 |
| $TiO_2$ | 0.835 | Ti | 0.588 | $Co_3O_4$ | 0.0038 | Co | 0.0042 |
| $P_2O_5$ | 0.204 | Px | 0.30 | PdO | 0.0054 | Pd | 0.0044 |
| MnO | 0.127 | Mn | 0.0733 | $TeO_2$ | 0.0053 | Te | 0.0035 |
| $SO_3$ | 0.0444 | Sx | 0.0807 | $Ga_2O_3$ | 0.0033 | Ga | 0.0027 |
| SrO | 0.0404 | Sr | 0.0318 | $WO_3$ | 0.0033 | W | 0.0026 |
| ZnO | 0.0218 | Zn | 0.0157 | PbO | 0.0034 | Pb | 0.0029 |
| $V_2O_5$ | 0.0217 | V | 0.0283 | $Sc_2O_3$ | 0.0014 | Sc | 0.0024 |
| Cl | 0.02 | Cl | 0.0116 | $RuO_4$ | 0.0014 | Ru | 0.0023 |

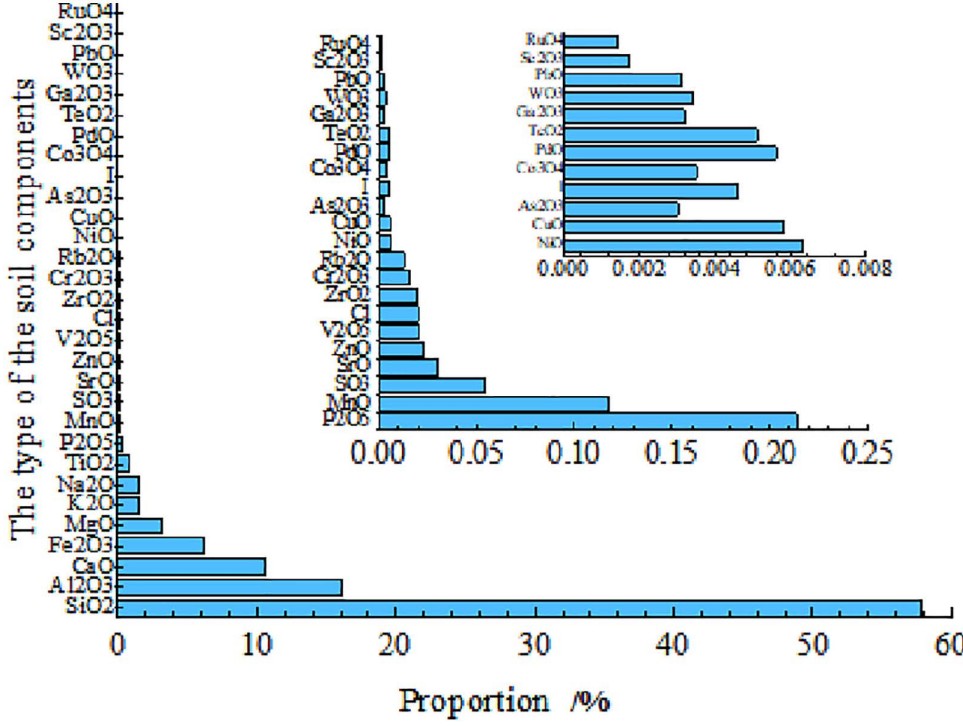

**Fig 4. Results of XRD analysis of test soil.**

### (2) Sodium methylsilicate

Sodium methylsilicate is an inorganic chemical compound known for its waterproof properties. It is extensively utilized in both everyday waterproofing applications and industrial protective measures. The waterproof properties are attributed to the unique chemical bonds formed by silanol groups (Si-OH) within the structure. These bonds can interact with silanol groups present in the material, leading to the formation of methyl groups (-CH3). As a result, this interaction imparts excellent hydrophobicity.

### (3) Styrene-acrylic emulsion

The styrene-acrylic emulsion utilized in the test is a milky white liquid with a blue hue, featuring a solid content ranging from 40% to 50%, and a viscosity between 80 and 200 mPa·s. It exhibits weak alkalinity, along with excellent adhesion and water resistance. This emulsion can serve as one of the additives for repair materials.

### (4) Waterborne polyurethane

The waterborne polyurethane utilized in this test is a one-component, anionic polyurethane emulsion. It appears as a milky white liquid in its unprocessed state and becomes colorless and transparent upon drying. The viscosity ranges from 800 to 1500 mPa·s, with a pH value between 6 and 8. This material exhibits excellent elasticity, wear resistance, and weatherability, making it suitable for use as one of the reinforcing agents in repair materials.

### (5) Urea and urease

The urea utilized in this test is analytically pure urea obtained from Tianjin No.3 Chemical Factory Co., Ltd. Its chemical composition is $H_2NCONH_2$, with a relative molecular weight of 60.06 and a purity of no less than 99%. This substance

appears as white crystals and is classified as an organic compound. Urea is easy to preserve and utilize. Urea solution can be catalyzed by urease to produce carbonate ions ($CO_3^{2-}$). These carbonate ions can then react with free calcium ions ($Ca^{2+}$) to form calcium carbonate, resulting in the precipitation of calcium carbonate, which enhances the strength of soil samples. Urease is synthesized in a manner that allows for its metabolism by specific organisms or plants, such as beans and fruits that are prevalent in our daily lives. It can be utilized to catalyze the decomposition of urea into ammonia ions and carbonate particles. The urease used in this study is manufactured by Shunxing Biotechnology Co., Ltd., with an effective content of no less than 99%. Using Whiffin's electrical conductivity method to assess urease activity, as derived from formula (6), the urease activity was measured at 1.76 μmol urea hydrolyzed/min.

$$U = 10 \times KA \tag{6}$$

Where U: Unit enzyme activity (μmolurea/min)

 K: Conductivity change, μS/min

 A: Specific coefficient (A = 11.11)

## 3. Experimental design and method

### 3.1. Experimental design

To ensure the test has broad applicability, the primary component of the repair material is silty clay. The waterproofing agents incorporated include sodium methylsilicate, styrene-acrylic emulsion, and waterborne polyurethane. Additionally, the water utilized consists of urease solution and urea solution. In conjunction with the commonly utilized quicklime content, quicklime is selected at concentrations of 7.5%, 15%, and 22.5% [27]. To facilitate a more effective comparison of the performance differences among various components, the dosages of the waterproofing agent are set at 3%, 5%, and 7%. Additionally, both the urease and urea solutions are maintained at a concentration of 1 mol/L. According to the literature, the moisture content of grouting materials typically ranges from 53% to 58%. Considering the characteristics of silty clay, a moisture content of 55% has been selected as the optimal level for this experiment to ensure adequate fluidity. The testing scheme is presented in Table 4 below:

Table 4. Test Scheme.

| Serial number | Quicklime | Admixture | Serial number | Quicklime | Admixture | Serial number | Quicklime | Admixture |
|---|---|---|---|---|---|---|---|---|
| 7.5% + J3 | 7.5% | 3% sodium methylsilicate | 15% + J3 | 15% | 3% sodium methylsilicate | 22.5% + J3 | 22.5% | 3% sodium methylsilicate |
| 7.5% + J5 | | 5% sodium methylsilicate | 15% + J5 | | 5% sodium methylsilicate | 22.5% + J5 | | 5% sodium methylsilicate |
| 7.5% + J7 | | 7% sodium methylsilicate | 15% + J7 | | 7% sodium methylsilicate | 22.5% + J7 | | 7% sodium methylsilicate |
| 7.5% + B3 | | 3% styrene-acrylic emulsion | 15% + B3 | | 3% styrene-acrylic emulsion | 22.5% + B3 | | 3% styrene-acrylic emulsion |
| 7.5% + B5 | | 5% styrene-acrylic emulsion | 15% + B5 | | 5% styrene-acrylic emulsion | 22.5% + B5 | | 5% styrene-acrylic emulsion |
| 7.5% + B7 | | 7% styrene-acrylic emulsion | 15% + B7 | | 7% styrene-acrylic emulsion | 22.5% + B7 | | 7% styrene-acrylic emulsion |
| 7.5% + JA3 | | 3% waterborne polyurethane | 15% + JA3 | | 3% waterborne polyurethane | 22.5% + JA3 | | 3% waterborne polyurethane |
| 7.5% + JA5 | | 5% waterborne polyurethane | 15% + JA5 | | 5% waterborne polyurethane | 22.5% + JA5 | | 5% waterborne polyurethane |
| 7.5% + JA7 | | 7% waterborne polyurethane | 15% + JA7 | | 7% waterborne polyurethane | 22.5% + JA7 | | 7% waterborne polyurethane |

## (1) Selection of urea and urease ratio

During the construction of ancient earthen city walls, glutinous rice paste was employed to improve the strength and durability of the earthen structures. However, the preparation process of glutinous rice pulp is quite cumbersome, which does not align with modern construction technologies. Additionally, its inherent fluidity is limited, leading to a reduction in the overall fluidity of repair materials. Through the research conducted by relevant scholars on MICP and EICP, it has been determined that utilizing microorganisms to facilitate calcium carbonate deposition represents an environmentally friendly soil solidification technology. This method not only minimizes environmental pollution but also demonstrates a high degree of compatibility with the original local city wall [9]. The addition of quicklime, despite the absence of $CaCl_2$-commonly utilized in their practices-can significantly enhance the compressive strength and permeability of the repaired samples. Furthermore, it contributes to an improvement in the physical and mechanical properties of the soil to a certain degree. In this experiment, although $CaCl_2$ is omitted, a substantial amount of quicklime has been incorporated alongside high water content. Quicklime reacts with water to produce calcium hydroxide ($Ca(OH)_2$), which also releases $Ca^{2+}$ ions. To enhance the overall performance of repair materials, a mixture of urea and urease is selected for incorporation into the water. During this reaction process, urease primarily serves a catalytic function throughout the entire reaction. The amount of urease added directly influences the completeness of the response. Under the catalytic action of urease, urea decomposes to yield $NH_3$ and $CO_2$. The $NH_3$ subsequently hydrolyzes in water to form $NH_4^+$ and $OH^-$, while $CO_2$ dissolves in water to generate $CO_3^{2-}$ and $H^+$. These products then interact with $Ca^{2+}$ present in the materials, resulting in the precipitation of $CaCO_3$. This process ultimately enhances the overall performance of grouting materials. The main reactions are as follows:

$$CO(NH_2)_2 + H_2O \rightarrow 2NH_3 + CO_2$$

$$2NH_3 + 2H_2O \rightarrow 2NH_4^+ + 2OH^-$$

$$CO_2 + H_2O \rightarrow HCO_3^- + H^+$$

$$HCO_3^- + H^+ + 2OH^- \rightarrow CO_3^{2-} + 2H_2O$$

$$Ca^{2+} + CO_3^{2-} \rightarrow CaCO_3 \downarrow$$

Excessive levels of urea can result in a reduction of urease activity [28]. To optimize the aforementioned reaction, we have determined that a urease concentration of 15 g/L and urea at 1 mol/L are commonly employed. It is important to note that the required water consumption varies depending on the quicklime content present. However, since the primary component of urea is ammonium salt, both the concentration and quality of urea are excessively high. Consequently, the content of ammonium salt in the solution also increases, leading to a higher salt concentration that can accelerate the deterioration of repair materials. Therefore, the water required for mixing is initially categorized into two groups: pure water and urea at ratios of 0:1 and 1:1. Experimental results indicate that the optimal mixing involves a group characterized by a specific ratio of pure water to urea. After 14 days of mixing, a noticeable alkaline phenomenon will manifest on the surface (Fig 5), which adversely affects both strength and durability. Group 1 did not exhibit any signs of alkali flooding. Consequently, this ratio was utilized in the subsequent experiments.

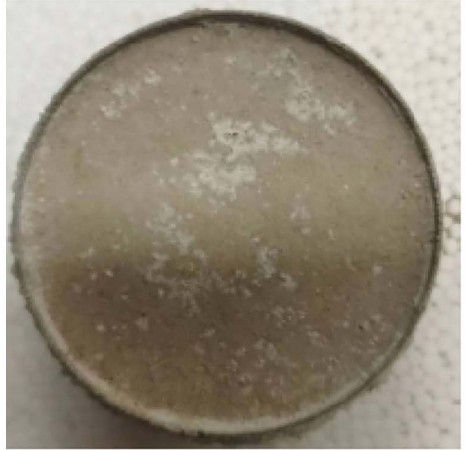 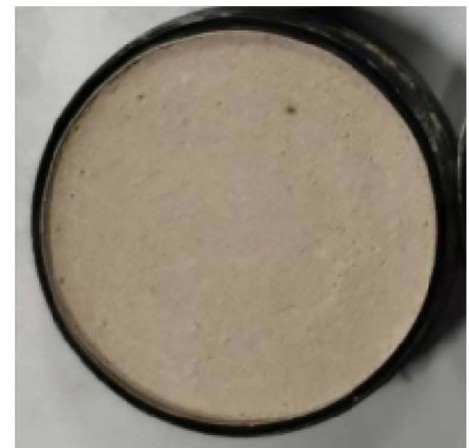

(a) 1:0 urea urea enzyme content.  (b) 1:1 urease content.

**Fig 5. Surface situation of different urease content.**

**(2) Selection of Quicklime Content**

Adding quicklime not only adheres to the principle of "repairing the old as the old" in the restoration of ancient buildings, but also mitigates drying shrinkage deformation that arises from water loss during the drying process. This is particularly important given the high moisture content of repair materials, which can adversely affect both their strength and overall effectiveness in achieving a successful restoration. As illustrated in Fig 6a, the left image depicts the group that did not incorporate quicklime. There is significant shrinkage surrounding the repair materials. In contrast, the right image represents the group with added quicklime, where no noticeable drying shrinkage can be observed around it. As illustrated in Fig 6b, the left image depicts the group that did not incorporate quicklime. There is significant shrinkage surrounding the repair materials. In contrast, the right image represents the group with added

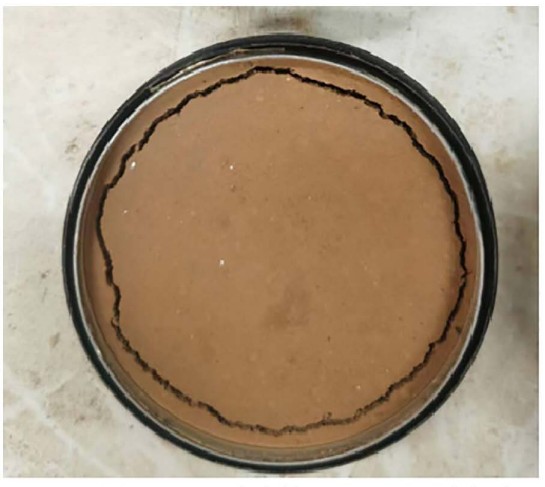

(a) No quicklime was added.  (b) Quicklime is added.

**Fig 6. Strength diagram of surface without quicklime and with lime added.**

quicklime, where no noticeable drying shrinkage can be observed around it [29]. The quicklime content utilized in this experiment, informed by the prevalent quicklime levels and local folklore regarding "1900 and 2800 quicklime soil," is set at 7.5%, 15%, and 22.5%.

## 3.2. Sample preparation

Prior to the experiment, silty clay collected near the earthen city wall will be sun-dried, crushed, sieved through a 2 mm mesh, and dried in an oven. Quicklime will then be added at varying proportions and mixed thoroughly. The mixture will be allowed to rest for 24 hours to ensure complete reaction between quicklime and soil. A solution of urease, urea, and water in a 1:1:2 ratio will be prepared. Subsequently, different types and concentrations of waterproofing agents will be incorporated into the solution and mixed uniformly. This solution will then be added to varying amounts of quicklime-soil mixtures and stirred for 3 minutes to ensure homogeneity. The resulting mixtures will be poured into standard molds, vibrated for 30 seconds to eliminate air bubbles and ensure uniform material distribution, and subsequently cast into specimens. Each group will consist of three parallel specimens. After preparation, the specimens will be cured for 7 days under controlled conditions. Following curing, surface strength tests, water absorption tests, post-water absorption surface strength tests, and water loss tests will be conducted on the specimen surfaces (Fig 7). Once the data has been analyzed, the optimal repair material ratio will be determined using the effectiveness coefficient method, and crack repair trials will be performed. Cracks will first undergo pre-treatment (e.g., moistening the surrounding area), followed by the injection of repair materials to ensure complete filling of the cracks. Upon completion of the repair, the specimens will be transferred to a curing chamber for an additional 7-day curing period. The curing chamber will maintain a temperature of $(20\pm2)°C$ and a relative humidity exceeding 95%. During this period, specimens will be misted daily to ensure continuous moisture retention without water accumulation.

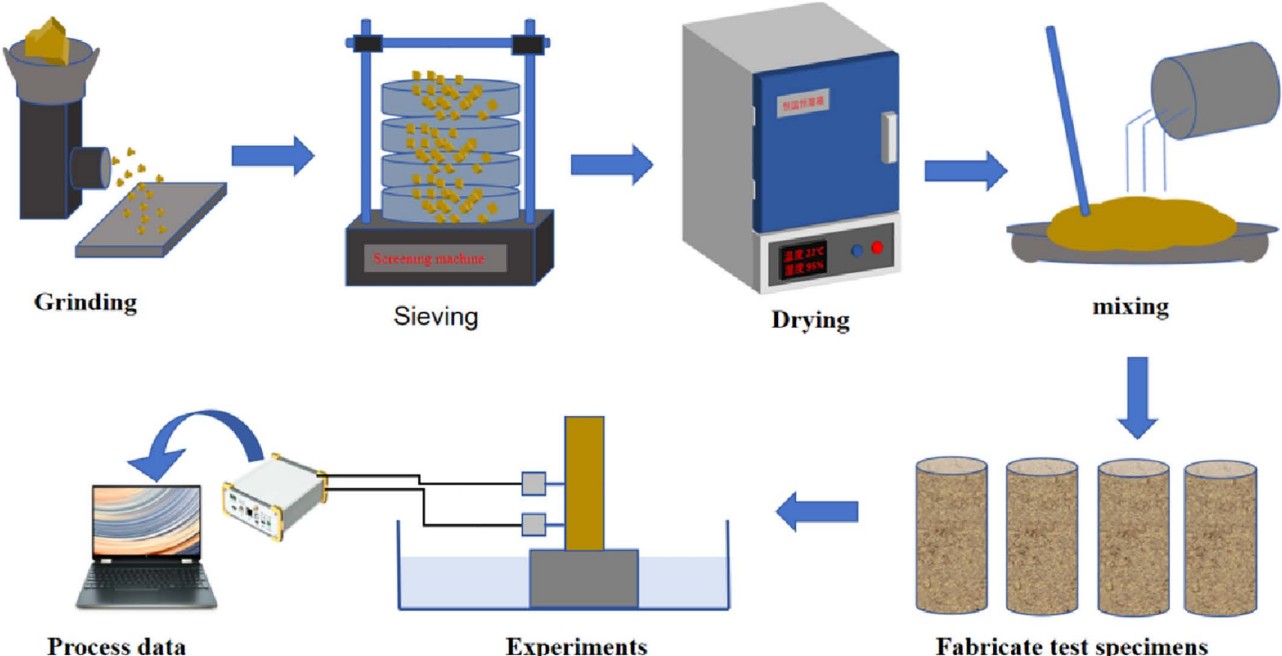

**Fig 7. Experimental workflow of crack repair materials.**

### 3.3. Test methods

To better investigate the properties of various repair materials, this paper conducts a series of tests including mortar consistency test, surface strength test, water absorption test, water loss test, and freeze-thaw cycle test following the repair process.

**(1) Consistency test of mortar**

Place the prepared repair material into the testing instrument, ensuring that it is positioned approximately 8 cm away from the mouth of the container. Use a tamping rod to vibrate both the center and edges of the mortar 25 times, followed by tapping the exterior 5–6 times. This process will ensure that the surface of the mortar is ultimately level and smooth. Place the prepared container in the designated position. Adjust the test cone so that it just makes contact with the surface of the material. Securely tighten the brake screw, ensuring that the lowest end of the rack side rod is in contact with the upper end of the slide rod. Verify whether the pointer returns to zero; if so, loosen the brake screw. After allowing a standing time of 10 seconds, retighten the screw and measure again at the point where the lower end of the rack side rod meets with the upper end of the slide rod. Record this measurement as it represents the consistency value of your repair material. To ensure data validity, conduct three measurements using slurry mixtures with identical ratios.

**(2) Surface strength test**

Place the prepared sample on the surface strength tester, ensuring that the probe makes contact with the sample's surface. Adjust the pressure gauge to zero and record both the position of the probe and the pressure value indicated by the press when the probe's position decreases by 1 cm. Each sample should be measured three times, and an average of these measurements should be calculated.

**(3) Water absorption test**

After weighing the cured samples, they were placed in a sink, and the mass changes at 3 hours, 6 hours, 12 hours, and 24 hours were recorded respectively. The ratio of mass changes to the dry weight of the samples was utilized as the water absorption index for the repair materials, with three horizontal samples included in each group.

**(4) Surface strength test after water absorption**

Remove the sample that has absorbed water for 24 hours and place it on the surface strength tester. Repeat the aforementioned surface strength testing procedure, and document its surface strength accordingly.

**(5) Freeze-thaw cycle test**

After repairing the simulated crack samples, we aim to assess the durability of the repaired materials in a real-world environment. Based on the temperature fluctuations observed in Kaifeng, we have established a freeze-thaw cycle temperature range from −10°C to 20°C. Each cycle is designed to last for 24 hours, comprising 12 hours of freezing followed by 12 hours of thawing. A total of 30 freeze-thaw cycles were conducted to simulate the extreme winter weather conditions in the Kaifeng area. Surface changes were recorded after each cycle to identify the most suitable material for repairing cracks in earthen structures.

## 4. Test results

### 4.1. Test results of repair materials with different mix ratios

**(1) Consistency test analysis of repairing material mortar**

The flowability of 27 types of repair materials was tested, taking into account the actual repair process. Given that gravity grouting is employed, the flowability significantly impacts the practical operability. If the flowability is limited, it

indicates that the material possesses exceptional performance but cannot be easily poured. It is evident from Fig 8 that as the content of quicklime increases, the overall consistency of high-flow restoration materials exhibits a general downward trend. With the addition of 3% styrene-acrylic emulsion, the consistency of restoration materials containing 7.5% quicklime is measured at 130 mm. In contrast, the consistency of restoration materials with a quicklime content of 22.5% is only 123 mm. This represents a decrease in consistency value by 7 mm, corresponding to a reduction rate of 5%. However, the consistency of the 3% waterborne polyurethane group decreased from 120 mm to 117 mm with an increase in quicklime content, resulting in a reduction of approximately 3%. This phenomenon can be attributed to the high water absorption capacity of quicklime itself, as well as the stoichiometric ratio of $Ca(OH)_2$ to water being 1:1. As the quicklime content increases, the amount of $Ca(OH)_2$ also rises correspondingly, necessitating a greater volume of water. Furthermore, it is essential to note that the reaction between quicklime and water to form $Ca(OH)_2$ is exothermic. However, the consistency of the 3% waterborne polyurethane group decreased from 120 mm to 117 mm with an increase in quicklime content, resulting in a reduction of approximately 3%. This phenomenon can be attributed to the high water absorption capacity of quicklime itself, as well as the stoichiometric ratio of $Ca(OH)_2$ to water being 1:1. As the quicklime content increases, the amount of $Ca(OH)_2$ also rises correspondingly, necessitating a greater volume of water. Furthermore, it is important to note that the reaction between quicklime and water to form $Ca(OH)_2$ is exothermic. As the quicklime content increases, the heat generated during hydration also rises. This phenomenon can result in a significant release of heat within a short period during mixing operations, leading to the evaporation of some water due to excessive temperatures. Consequently, this may cause a decline in the consistency of repair materials.

By examining the overall consistency of various waterproofing agents (Fig 8), it is evident that the sodium methylsilicate group exhibits the highest level of consistency, whereas waterborne polyurethane demonstrates the lowest. This distinction arises from the fact that sodium methylsilicate is a liquid additive, whereas styrene-acrylic emulsion and waterborne polyurethane are classified as white emulsion liquids. Although waterborne polyurethanes are readily soluble in water, they exhibit the highest viscosity among similar materials, reaching values between 800 and 1500 mPa·s. The viscosity of a substance is directly proportional to its viscosity coefficient; thus, higher viscosity results in lower fluidity when mixed with soil. Styrene-acrylic emulsion possesses a certain level of viscosity, yet it can modify the interfacial bonding between

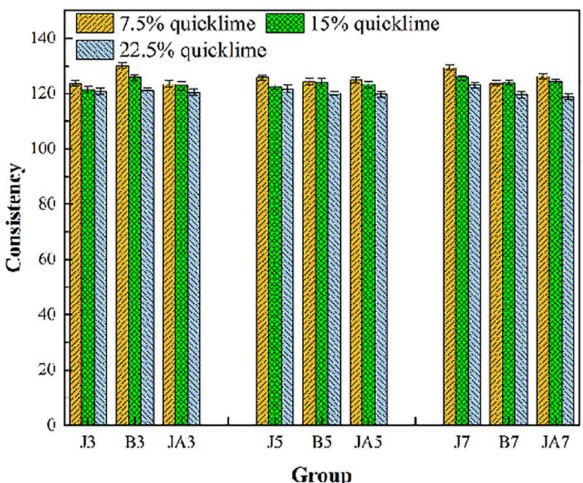

(a) Consistency according to the admixture group.

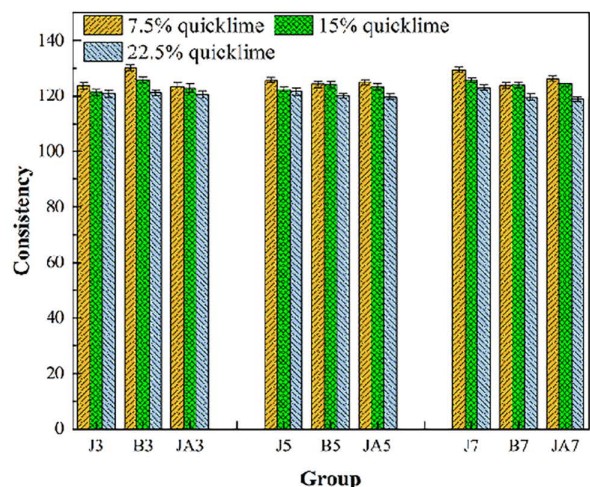

(b) Consistency grouped by content.

**Fig 8. Consistency of different groups.**

particles, thereby enhancing consistency to some degree. It is evident that the viscosity of waterproofing agents significantly influences the consistency of the repair material itself.

For the low quicklime production group, the consistency is significantly influenced by the content of waterproofing agents. As the quicklime content increases, the heat generated from both quicklime reactions and chemical processes exerts a greater impact on the consistency of repair materials, gradually assuming a dominant role. This is due to the relatively small proportion of low-yield quicklime in the soil, which is significantly influenced by the inherent properties of the waterproofing agent. Additionally, sodium methylsilicate impedes its hydration reaction during the interaction with quicklime, thereby reducing the degree of response and minimizing water loss. This process ultimately enhances consistency and improves fluidity. For quicklime contents of 15% and 22.5%, the amount of quicklime is greater, while the proportion of the waterproofing agent added is lower than that of quicklime. Consequently, in groups with high quicklime content, the effect of the waterproofing agent on consistency is minimal.

**(2) Experimental analysis of the surface strength of repair materials**

For the cracked soil wall, high-flow repair materials primarily address the cracks, with a focus on evaluating surface strength. As illustrated in Fig 9, the surface strength of high-fluid repair materials incorporating styrene-acrylic emulsion and waterborne polyurethane is notably higher. In contrast, the overall strength of the sodium methylsilicate group is comparatively lower. For methylsilicate, the surface strength exhibits a decreasing trend with an increase in sodium methylsilicate content, while it shows an increasing trend with higher quicklime content. When sodium methylsilicate is added at a concentration of 7%, the quicklime content has minimal impact on the overall strength. Under conditions where sodium methylsilicate is present at 3%, the effect of quicklime content on surface strength becomes most pronounced. Specifically, at 7.5% quicklime content, the strength reaches 16.3, and at 22.5% quicklime content, it increases to 46.5—representing a remarkable enhancement of 180%. This group also demonstrates the highest surface strength within the sodium methylsilicate category. For styrene-acrylic emulsion and waterborne polyurethane, the surface strength exhibited a trend of initially increasing and then decreasing with varying quicklime content at a constant concentration. The lowest strength was observed at 7.5% quicklime content, while the highest strength occurred at 15% quicklime content. When the content

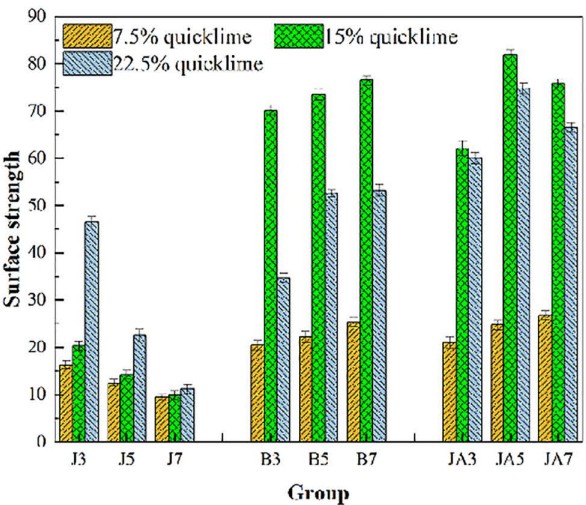

(a) Surface strength grouped by admixture.

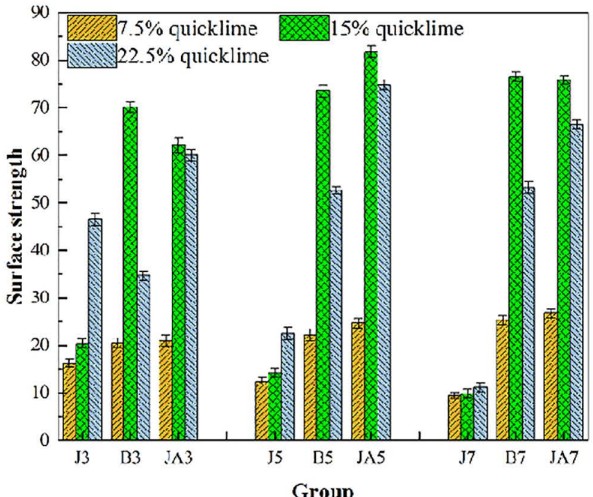

(b) Surface strength grouped by content.

**Fig 9. Surface strength of different groups.**

of styrene-acrylic emulsion is set at 5%, the surface strength of high fluid repair materials influenced by quicklime exhibits a significant increase. Specifically, it rises from 22.2% with a quicklime content of 7.5% to 73.5% with a quicklime content of 15%, representing an overall enhancement of 230%. When the content of waterborne polyurethane is 5%, the surface strength of the repair material with 7.5% quicklime content measures 24.7, while that with 15% quicklime content reaches 81.8. This represents an increase of nearly 230%. Although the content of styrene-acrylic emulsion and waterborne polyurethane remains constant, the surface strength initially increases with the addition of quicklime before subsequently decreasing. Notably, the decline in surface strength for styrene-acrylic emulsion is more pronounced, whereas waterborne polyurethane exhibits only a slight decrease.

The differences in the properties of the three materials can be attributed to their distinct repair principles, despite all of them enhancing the surface strength of the repair materials. Waterborne polyurethane contains a prepolymer known as an active isocyanate group (-NCO). When this prepolymer comes into contact with water during mixing or with moisture in the air, it rapidly reacts with water to form a high molecular weight polymer. This polymer can effectively fill the voids between soil particles and encapsulate them, thereby enhancing the overall strength of the soil [30]. However, as the content of quicklime and waterborne polyurethane increases, the amount of water consumed by the quicklime itself also rises. This results in an incomplete reaction of the active isocyanate groups (-NCO). Furthermore, the reacted active isocyanate groups may encapsulate both future reactive quicklime and existing reactive components, leading to mutual inhibition between these two reactions. However, with a quicklime content of 7.5%, although the water supply is adequate, the amount of waterborne polyurethane present is reduced, resulting in a lower enhancement of strength. The styrene-acrylic emulsion forms a polymer film that interweaves with the compounds produced by quicklime, creating a dense interwoven network structure. This configuration significantly enhances the integrity of the repair material, improves its mechanical properties, and increases the surface strength of the repair material. However, when the quicklime content is excessively high, its inherent expansion can lead to the formation of micro-cracks. This phenomenon compromises the integrity of the network structure established by styrene-acrylic emulsion, resulting in a strength that is inferior to that observed with 15% quicklime [31].

Sodium methylsilicate solution permeates the voids between silicate gel particles present in the solution, effectively binding calcium carbonate and calcium hydroxide produced in the soil into a cohesive mass. This process enhances the overall strength of the material. However, as the content increases, the amounts of carbon dioxide and $H_2O$ available in the repair material for reaction with sodium methylsilicate diminish. Consequently, sodium methylsilicate is unable to generate a sufficient film to fill additional gaps. This phenomenon is a primary factor contributing to the gradual decrease in the strength of the sodium methylsilicate group [32]. After the quicklime is hydrated with water, the calcium ions ($Ca^{2+}$) from calcium hydroxide will react with carbonate ions ($CO_3^{2-}$) generated through the carbonization of sodium methylsilicate and those produced by the mineralization of urease. This reaction leads to the formation of calcium carbonate ($CaCO_3$), thereby enhancing the mechanical properties of remediation materials.

### 4.2. Test results of hydraulic properties of restoration materials

### (1) Water absorption test analysis

Currently, the primary cause of damage to earthen sites is the varied and repeated effects of water. Additionally, a significant amount of water is incorporated into high-flow restoration materials themselves, making the water absorption capacity of these materials a crucial parameter. As illustrated in Fig 10, for the group with 7.5% quicklime content, the water resistance of the waterborne polyurethane exhibits superior performance, followed by sodium methylsilicate, and lastly styrene-acrylate emulsion. The water absorption of 5% waterborne polyurethane over 24 hours is the lowest, recorded at 3.9%. In contrast, the water absorption of a 3% styrene-acrylate emulsion during the same timeframe reaches 20%, which is five times greater than that of the 5% waterborne polyurethane. As the quicklime content increases, there

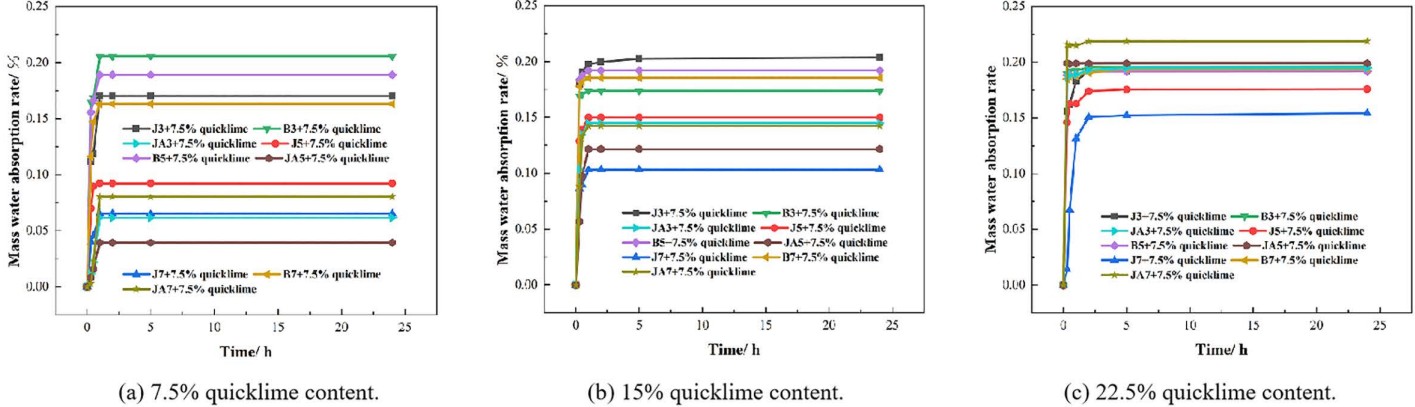

(a) 7.5% quicklime content.　(b) 15% quicklime content.　(c) 22.5% quicklime content.

**Fig 10. Water absorption of different groups of mass.**

is a noticeable upward trend in overall water absorption. Specifically, the water absorption for the group with 22.5% quicklime content ranges from 15% to 22%. Under a quicklime content of 22.5%, both 5% and 7% waterborne polyurethane exhibit poor waterproof properties, achieving only 22%. This performance is notably higher than that observed at a quicklime content of 7.5%, respectively. Under the duplicate quicklime content, the water absorption of sodium methylsilicate decreases as the concentration of sodium methylsilicate increases. At a parameter level of 7.5% quicklime, the water absorption of high fluid restoration material containing 3% sodium methylsilicate is measured at 17%. In contrast, when the sodium methylsilicate content is reduced to 7%, the water absorption drops significantly to only 6%, representing an 11% decrease compared to that observed with a sodium methylsilicate content of 7%. As the quicklime content increases, the water absorption of sodium methylsilicate gradually rises. When the sodium methylsilicate content is 5%, the water absorption of the repair material is measured at 9.22% with a quicklime content of 7.5%. In contrast, when the quicklime content increases to 22.5%, the water absorption of the repair material rises to 17.3%, resulting in an increase of 6%. However, the styrene-acrylic emulsion remains unaffected by both quicklime content and its own concentration, which is consistently maintained at approximately 18%. The water absorption of 27 groups of high fluidity remediation materials exhibited a consistent variation with the increasing quicklime content and self-content; however, all values remained below 35% compared to the plain soil group. This indicates that the incorporation of admixtures into high fluidity repair materials can significantly enhance their waterproof performance.

The waterproofing methods employed by the three materials differ significantly. Sodium methylsilicate, an inorganic compound, forms a hydrophobic layer of polysiloxane film on the surface of the repair material particles. The hydrophobic layer will efficiently isolate and inhibit the ingress of water, thereby minimizing water absorption [21]. Waterborne polyurethane and styrene-acrylate emulsion are classified as organic materials. Waterborne polyurethane chemically bonds linear polyurethane macromolecules together, thereby imparting a certain degree of waterproof effectiveness. The styrene-acrylic emulsions creates a robust hydrophobic film between particles, thereby enhancing the waterproof properties of repair materials [33].

Sodium methylsilicate reacts with atmospheric carbon dioxide to produce silicic acid ($H_2SiO_3$) and carbonate ions ($CO_3^{2-}$) or bicarbonate ions ($HCO_3^-$). This reaction leads to the formation of an insoluble silicic acid gel on the surfaces of soil particles and within cracks, thereby enhancing the hydraulic properties of soil particles adjacent to fissures in urban earthen walls. Compared to styrene-acrylic emulsion, sodium methylsilicate exhibits a broader penetration range during repair processes and significantly enhances the waterproof performance of soil on both sides of cracks. The hydrophobic layer of the polysilicane film, formed by sodium methylsilicate, requires a combination of $CO_2$ and moisture from the air.

Similarly, the carbonization process of quicklime in the atmosphere also necessitates both $CO_2$ and humidity. When the quicklime content is low, the required amounts of $CO_2$ and humidity are reduced. Under these conditions, sodium methylsilicate can react with sufficient $CO_2$ to form a hydrophobic layer of polysiloxane film. The quantity of sodium methylsilicate directly influences the production of films and enhances the waterproofing effect. However, an excessive quicklime content can lead to competition between quicklime and sodium methylsilicate for $CO_2$ and moisture present in the air [34]. This competition ultimately results in a reduction in the formation of silica films. When quicklime undergoes carbonization to produce calcium carbonate, its volume expands. An excessive amount of quicklime may lead to the formation of micro-cracks in the repair material. Water can infiltrate through these micro-cracks, which is a primary factor contributing to the increased water absorption of the repair material as quicklime content rises.

**(2) Experimental analysis of surface strength after water absorption**

For high-flow repair materials, the strength following water absorption significantly impacts both the effectiveness of the repair and its durability post-absorption. If the surface strength following water absorption is insufficient, it will deteriorate rapidly after exposure to snow and rainfall, leading to unsatisfactory repair outcomes. As illustrated in Fig 11, the strength diminishes following water absorption; however, the surface strength of both the styrene-acrylic emulsion group and the waterborne polyurethane group remains elevated after exposure to moisture. For the sodium methylsilicate group, the surface strength of the 22.5% quicklime content group exhibits the highest value after water absorption at a 3% content level. However, when compared to its pre-water absorption state, this strength experiences a significant decrease of 57%, indicating a substantial decline. Conversely, under both 7% and 7.5% quicklime content levels, the surface strength shows the smallest reduction—from 9.5 to 7.1—resulting in an approximate decrease of only 26%. Nonetheless, it is important to note that the initial strength remains relatively low. For the sodium methylsilicate group, the surface strength of the 22.5% quicklime content group exhibits the highest value after water absorption at a 3% content level. However, when compared to its pre-water absorption state, this strength experiences a significant decrease of 57%, indicating a substantial decline. Conversely, under both 7% and 7.5% quicklime content levels, the surface strength shows the smallest reduction—from 9.5 to 7.1—resulting in an approximate decrease of only 26%. Nonetheless, it is important to note that the initial strength remains relatively low. For the waterborne polyurethane group, an increase in quicklime content correlates with a more significant reduction in the surface strength of the repair materials. Under a quicklime content of 7.5%, the surface

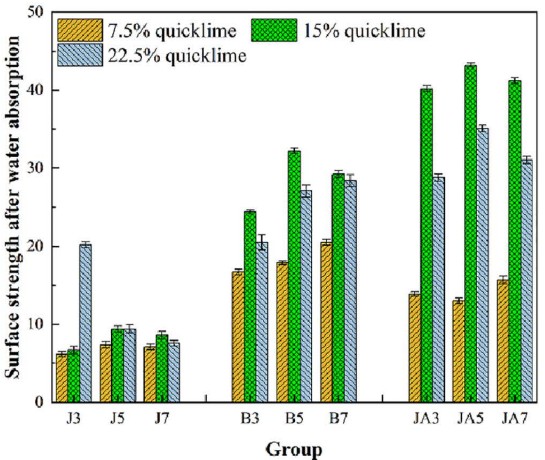

(a) Surface strength after water absorption grouped by admixture.

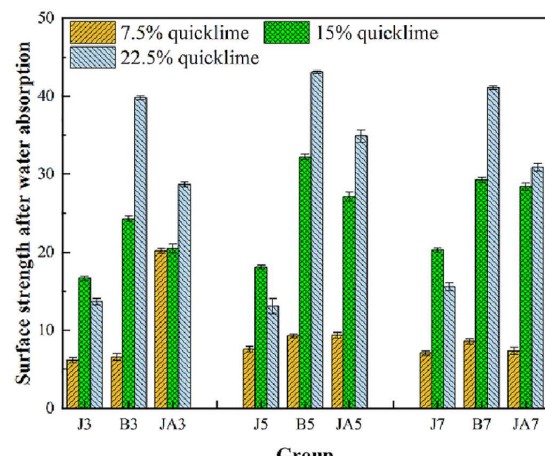

(b) Surface strength after water absorption grouped by content.

**Fig 11. Surface strength after water absorption in different groups.**

strength of three types of waterborne polyurethane decreased by 33%, 48%, and 42% respectively, marking this as the lowest observed value. For a quicklime content of 22.5%, the surface strength of three types of waterborne polyurethane decreased by 53%, 54%, and 57% respectively.

The primary reason for this phenomenon is the variation in waterproof performance among repair materials, which leads to differing levels of water absorption. The strength of repair materials exhibiting higher water absorption significantly decreases, whereas the strength of those with lower water absorption diminishes only slightly. This phenomenon occurs due to the continuous increase in the moisture content of the repair material as water absorption rises. This escalation brings the material closer to its liquid limit, which subsequently compromises its integrity and diminishes the surface strength of the repair material. For the sodium methylsilicate group, a hydrophobic film is formed on the surface of the particles. Once the hydrophobic film is compromised, dry soil will rapidly absorb water, allowing it to permeate the internal voids of the soil. This process will subsequently diminish its surface strength [35]. For styrene-acrylic emulsion, a hydrophobic film is formed between the particles. However, with the continuous infiltration of water, this hydrophobic film becomes compromised, resulting in increased water absorption and a reduction in the strength of the repair materials. Under a quicklime content of less than 15%, the reduction in surface strength of the three styrene-acrylic emulsions following water absorption is most pronounced. This phenomenon can be attributed to the fact that the polymer film formed before water absorption serves as the primary factor contributing to the increase in surface strength. However, the ongoing infiltration of water into the soil compromises the integrity of the polymer biofilm, resulting in a reduction in the surface strength of the repair materials. At this stage, the surface strength of the supporting repair materials is primarily derived from quicklime. Consequently, the strength remains relatively consistent when the quicklime content is uniform. However, the excessive increase in strength before water absorption leads to a more pronounced reduction in surface strength following water absorption. In the case of waterborne polyurethane, the adhesion properties of the polymer contribute to an increase in particle surface roughness, which enhances friction between particles. This results in improved sample strength and overall stability [36]. However, the material continuously absorbs water, resulting in a reduction of friction between particles and a subsequent decrease in strength. The reason that the addition of more quicklime leads to a greater decline in the surface strength of the repair material is due to the continuous absorption of water. As unreacted quicklime continues to react, this process results in volume expansion and the formation of micro-cracks, which ultimately contribute to a further reduction in strength.

### (3) Analysis of water loss test

For restoration materials, the potential impact on the performance of earthen sites post-restoration is a crucial consideration. Afterward, if the air permeability of the restoration material is inadequate, the moisture present in the earthen site will not evaporate properly following restoration. This can lead to stress concentration on both the surface of the original soil and the restoration material. Consequently, not only will this fail to achieve the desired restoration effect, but it may also accelerate deterioration. Therefore, the water permeability of the repair materials was investigated. As illustrated in Fig 12, restoration materials containing 7.5% and 15% quicklime content exhibit a complete loss of water within 168 hours, whereas those with a quicklime content of 22.5% require up to 192 hours to achieve complete desiccation. However, the water loss of 27 groups of high-flow repair materials is largely consistent during the initial 24–36 hours when subjected to the duplicate quicklime content. This indicates that, after reaching saturation in water absorption, the initial rate of water loss is primarily influenced by the quicklime content rather than any additives used. However, beyond 36 hours, the rate of water loss for these materials begins to vary. For the 7.5% quicklime parameter group, the water loss rate of the styrene-acrylic emulsion is observed to be the slowest. Specifically, the water loss rate of the styrene-acrylic emulsion with a 7% content reaches only 78% over a period of 132 hours. In contrast, the 5% waterborne polyurethane group exhibits the highest water loss rate, achieving a remarkable 97% within the same timeframe, effectively nearing a dry state. This indicates that at low quicklime content, both waterborne polyurethane and sodium methylsilicate demonstrate superior

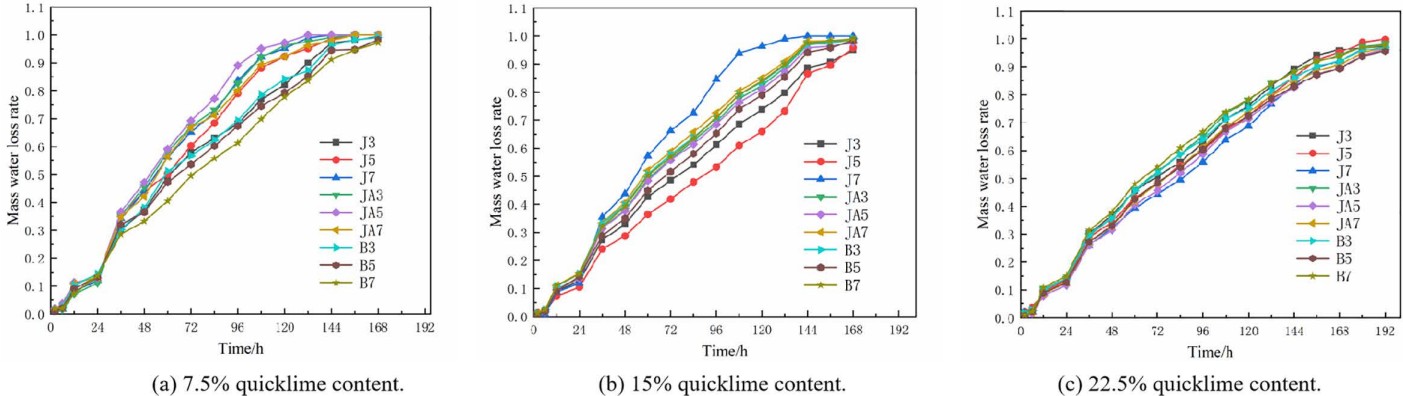

(a) 7.5% quicklime content. (b) 15% quicklime content. (c) 22.5% quicklime content.

**Fig 12. Mass water loss rate of different groups.**

water permeability compared to styrene-acrylic emulsion. For the group with 15% quicklime content, the most significant variation is observed in the sodium methylsilicate category. The formulation containing 7% sodium methylsilicate demonstrated exceptional water loss performance, achieving a remarkable rate of 96% within approximately 120 hours. In contrast, both the 3% and 5% sodium methylsilicate formulations exhibited markedly lower water loss rates of 65% and 59%, respectively, over the same period of time. This phenomenon occurs due to the increased concentration of waterproofing agents and quicklime, which necessitates the presence of water and carbon dioxide for the formation of new compounds. This situation can result in an inadequate reaction of sodium methylsilicate, leading to a reduced formation of silicon oxide films, uneven surface distribution, and diminished resistance to water evaporation. Consequently, the rate of water evaporation in the 7% sodium methylsilicate group is accelerated. For the group with 22.5% quicklime content, the trend of the water loss curve is generally consistent across samples. Notably, the groups containing 5% styrene-acrylic emulsion and 5% waterborne polyurethane exhibit poorer water loss performance. This can be attributed to an optimal admixture-to-quicklime ratio in these formulations. From the analysis of surface strength, it is evident that both groups exhibit a higher surface strength. This can be attributed to their relatively dense internal structure, which hinders the evaporation of water from within and consequently reduces water loss. However, the duration of water loss in the group with 22.5% quicklime content is longer than that observed in the groups with 7.5% and 15% quicklime content. This phenomenon can be attributed to the inherent properties of quicklime itself. As the quicklime content increases, it alters both the pore size and distribution within the soil, resulting in reduced porosity. Consequently, this makes it more challenging for water to evaporate, ultimately leading to a decrease in the overall rate of water loss [36].

## 4.3. Test of the actual joint filling effect

### 4.3.1. Mix proportion analysis of grouting repair materials.

**(1) Efficiency coefficient method**

Due to the varying compositions of different repair materials, each of the 27 groups exhibits distinct characteristics. The consistency of the group comprising 7.5% quicklime and 7% sodium methylsilicate is notably high, while the water absorption in the group containing 7.5% quicklime and 5% polyurethane is relatively low. Additionally, the surface strength of the group with 15% quicklime and 5% waterborne polyurethane demonstrates commendable performance. Therefore, to achieve a more effective balance of the comprehensive properties across each group of materials, the overall characteristics of 27 groups of materials are assessed using the efficacy coefficient method. The efficacy coefficient method is a

comprehensive analytical approach utilized for evaluating multiple indices. For each evaluation index, both a satisfactory value and an unsatisfactory value are established; the satisfactory value serves as the upper limit of evaluation, while the unsatisfactory value represents the lower threshold of assessment. The satisfaction level of each index in relation to the overall evaluation is calculated to determine the functional coefficient value for each index. Subsequently, a weighted comprehensive evaluation value is derived through appropriate weighting, thereby taking into account the overall situation of each group [37]. Through this approach, we can establish various efficacy coefficients based on the specific characteristics of each index according to the following standards. There are four categories of variables: huge variables indicate that a higher numerical value of this index corresponds to a more favorable outcome, whereas minimal variables exhibit the opposite relationship. There are also stable variables, which refer to target values that remain fixed. The last category is interval variables, indicating that the target value falls within a specific range, and the single coefficient is maximized. Given that the text aligns with the parameters of non-interval and stable variables, we have selected the other two types of variables.

The formula is as follows:

$$g_{1i} = \begin{cases} \dfrac{x_i - x_{ni}}{x_{yi} - x_{ni}} \times 40 + 60, & x_i < x_{yi} \\ 100, & x_i \geq x_{yi} \end{cases} \tag{7}$$

$$g_{1i} = \begin{cases} \dfrac{x_i - x_{ni}}{x_{yi} - x_{ni}} \times 40 + 60, & x_i > x_{yi} \\ 100, & x_i \leq x_{yi} \end{cases} \tag{8}$$

Among them, the i-th huge data evaluation index is $g_{1i}$; $X_i$ is the actual value of the i-th (i = 1, 2,..., m) evaluation index; $x_{yi}$ is the satisfactory value of the i-th evaluation index; $X_{ni}$ is the disallowed value of the i-th evaluation index. With the definition of a single efficacy coefficient, according to the actual situation, the weighting coefficient is comprehensively considered, and the final total efficacy coefficient is obtained by setting different weights. The formula is as follows:

$$G = \sum_{i}^{m} (g_i w_i) \tag{9}$$

Where: G is the total efficacy coefficient value of the evaluated object; $g_i$ is the single efficacy coefficient value of the i-th evaluation index; $w_i$ is the weight coefficient of the i-th evaluation index.

**(2) Analysis of the efficacy coefficient of grouting repair materials**

In this experiment, the efficacy coefficients of the five evaluation indexes are multiplied by distinct weight coefficients to reflect their relative importance, as shown in Table 5. In practice, based on the specific conditions of Kaifeng City, the weights for consistency and mass water absorption rate are set at 30%, the weight for mass water loss rate is set at 20%, and the remaining indices are assigned a weight of 10% each. Set allowable and non-allowable values for each metric. Given that greater surface strength, strength after water absorption, consistency, and mass water loss rate indicate superior performance of high-flow repair materials, huge variable values are adopted for these metrics. A smaller mass water absorption index indicates better performance of high-flow repair materials; therefore, minimal variable values are adopted for this metric. Satisfactory values for surface strength, strength after water absorption, consistency, and 96H mass water loss rate are determined to be 160% of the plain soil test values, specifically 32, 16, 196.8, and 100%, respectively. The impermissible values are set at 40% of the corresponding satisfactory values, namely 8, 4, 49.2, and 27%. The acceptable

**Table 5.** Five indicators of different groups of tests.

| Group | Surface strength | Mass water absorption | Consistency | Strength after water absorption | Water loss rate |
|---|---|---|---|---|---|
| 7.5%+J3 | 16.25 | 17.01% | 123.66 | 12.85 | 68.15% |
| 7.5%+J5 | 12.45 | 9.22% | 125.66 | 13.3 | 79.16% |
| 7.5%+J7 | 9.5 | 6.53% | 129.44 | 10 | 83.68% |
| 7.5%+B3 | 20.5 | 18.90% | 130.12 | 16.7 | 82.78% |
| 7.5%+B5 | 22.2 | 18.31% | 124.24 | 17.9 | 89.12% |
| 7.5%+B7 | 25.3 | 20.55% | 123.82 | 20.5 | 80.15% |
| 7.5%+JA3 | 21.0 | 6.16% | 123.42 | 21 | 69.35% |
| 7.5%+JA5 | 24.7 | 3.93% | 124.91 | 17.7 | 67.45% |
| 7.5%+JA7 | 26.8 | 8.04% | 126.25 | 26.8 | 61.35% |
| 15%+J3 | 14.4 | 19.76% | 121.31 | 6.7 | 61.26% |
| 15%+J5 | 13.5 | 15.00% | 122.4 | 9.4 | 53.33% |
| 15%+J7 | 9.9 | 10.33% | 125.65 | 8.6 | 84.60% |
| 15%+B3 | 76.2 | 17.37% | 125.74 | 24.5 | 69.41% |
| 15%+B5 | 73.5 | 19.19% | 124.06 | 25.1 | 68.55% |
| 15%+B7 | 70.1 | 18.55% | 123.87 | 26.4 | 72.62% |
| 15%+JA3 | 62.1 | 14.48% | 122.75 | 18.8 | 70.94% |
| 15%+JA5 | 81.8 | 12.16% | 123.24 | 35.1 | 65.30% |
| 15%+JA7 | 75.8 | 14.24% | 124.3 | 23.1 | 70.96% |
| 22.5%+J3 | 46.5 | 19.12% | 120.8 | 20.2 | 62.88% |
| 22.5%+J5 | 22.5 | 17.30% | 121.59 | 9.4 | 60.04% |
| 22.5%+J7 | 11.2 | 16.09% | 122.96 | 7.6 | 55.96% |
| 22.5%+B3 | 34.7 | 19.04% | 121.08 | 20.5 | 65.37% |
| 22.5%+B5 | 52.6 | 19.52% | 120.1 | 27.1 | 58.77% |
| 22.5%+B7 | 53.2 | 19.15% | 119.57 | 28.4 | 61.26% |
| 22.5%+JA3 | 60.03 | 19.27% | 120.47 | 28.8 | 64.06% |
| 22.5%+JA5 | 74.8 | 20.19% | 119.67 | 35.1 | 60.49% |
| 22.5%+JA7 | 66.5 | 21.85% | 118.83 | 31.1 | 66.86% |

value for mass water absorption is defined as 40% of the body test value, equating to 14%, while the unsatisfactory value is set at 160% of the test value, corresponding to 56%. By applying the aforementioned formula, the individual functional coefficients of various indices can be derived. Subsequently, by multiplying these coefficients by their corresponding weight coefficients, the total efficacy coefficients for different restoration materials can be obtained, which may then be utilized to evaluate each restoration material. The specific calculation values are presented in Table 6. According to the results, the mixture of 7.5% quicklime and 7% sodium methyl acid group exhibits the highest total efficacy coefficient, which is 96.25. This indicates that, under this ratio, the high-fluidity repair material demonstrates superior overall performance, as well as enhanced mechanical and waterproof properties.

### 4.3.2. Analysis of freeze-thaw cycle test results for crack-sealing specimens.

**(1) Study on the repairing operation of the sample**

The Kaifeng city wall is a historically protected cultural relic. To more effectively evaluate the specific performance of high-flow restoration materials during restoration efforts, cracks were artificially created in the laboratory by simulating the composition of the soil from the Kaifeng city wall. The specific procedure is illustrated in Fig 13. below. The comprehensive indices of 27 groups of high-fluidity restoration materials were calculated using the efficacy coefficient method, and

**Table 6. Parameters of five indexes after transformation in different groups of tests.**

| Group | Surface strength | Mass water absorption | Consistency | Strength after water absorption | Water loss rate | Total efficacy coefficient |
|---|---|---|---|---|---|---|
| 7.5% + J3 | 73.75 | 97.13 | 80.18 | 89.50 | 93.73 | 88.26 |
| 7.5% + J5 | 67.42 | 100.00 | 80.72 | 91.00 | 85.47 | 87.15 |
| 7.5% + J7 | 62.50 | 100.00 | 100.00 | 100.00 | 100.00 | 96.25 |
| 7.5% + B3 | 80.83 | 95.33 | 81.93 | 100.00 | 87.25 | 88.71 |
| 7.5% + B5 | 83.67 | 95.89 | 80.34 | 100.00 | 90.35 | 89.31 |
| 7.5% + B7 | 88.83 | 93.76 | 80.22 | 100.00 | 85.96 | 88.27 |
| 7.5% + JA3 | 81.67 | 100.00 | 80.11 | 100.00 | 80.66 | 88.33 |
| 7.5% + JA5 | 87.83 | 100.00 | 80.52 | 100.00 | 79.73 | 88.88 |
| 7.5% + JA7 | 91.33 | 100.00 | 80.88 | 100.00 | 76.74 | 88.75 |
| 15% + J3 | 70.67 | 94.51 | 79.54 | 69.00 | 76.70 | 81.52 |
| 15% + J5 | 68.33 | 99.05 | 79.84 | 78.00 | 72.81 | 82.86 |
| 15% + J7 | 63.17 | 100.00 | 80.72 | 75.33 | 88.14 | 85.69 |
| 15% + B3 | 100.00 | 96.79 | 80.74 | 100.00 | 80.69 | 89.40 |
| 15% + B5 | 100.00 | 95.06 | 80.29 | 100.00 | 80.27 | 88.66 |
| 15% + B7 | 100.00 | 95.67 | 80.24 | 100.00 | 82.26 | 89.22 |
| 15% + JA3 | 100.00 | 99.54 | 79.93 | 100.00 | 81.44 | 90.13 |
| 15% + JA5 | 100.00 | 100.00 | 80.07 | 100.00 | 78.68 | 89.76 |
| 15% + JA7 | 100.00 | 99.77 | 80.35 | 100.00 | 81.45 | 90.33 |
| 22.5% + J3 | 100.00 | 95.13 | 79.40 | 100.00 | 77.49 | 87.86 |
| 22.5% + J5 | 84.17 | 96.85 | 79.62 | 78.00 | 76.10 | 84.38 |
| 22.5% + J7 | 65.33 | 98.01 | 79.99 | 72.00 | 74.10 | 81.95 |
| 22.5% + B3 | 100.00 | 95.20 | 79.48 | 100.00 | 78.71 | 88.15 |
| 22.5% + B5 | 100.00 | 94.74 | 79.21 | 100.00 | 75.47 | 87.28 |
| 22.5% + B7 | 100.00 | 95.10 | 79.07 | 100.00 | 76.70 | 87.59 |
| 22.5% + JA3 | 100.00 | 94.98 | 79.31 | 100.00 | 78.07 | 87.90 |
| 22.5% + JA5 | 100.00 | 94.10 | 79.10 | 100.00 | 76.32 | 87.22 |
| 22.5% + JA7 | 100.00 | 92.52 | 78.87 | 100.00 | 79.44 | 87.31 |

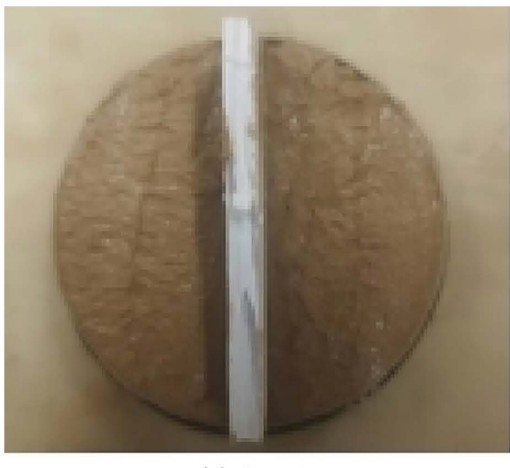
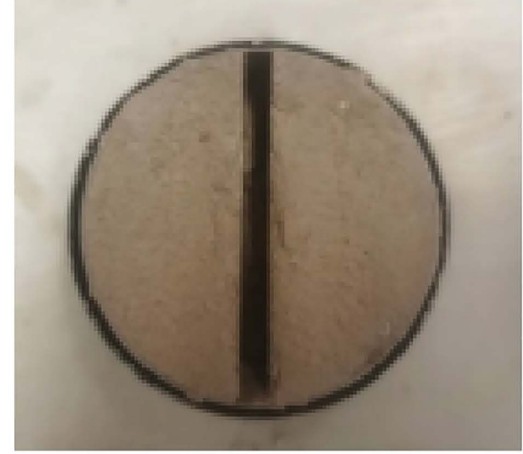

(a) Step1          (b) Step2

**Fig 13. Making diagram of simulating the cracks in Kaifeng city wall.**

the group with higher comprehensive index was selected for testing, namely the 7.5% quicklime + 7% sodium methylsili-cate group. The mixed high-fluidity repair material is injected into the crack, followed by the conduct of water absorption tests and freeze-thaw cycles to investigate the actual performance of the high-fluidity material within specific cracks. There are three groups, each containing three parallel samples.

After the high-flow repair material is fully injected into the crack, it is left to cure, and its performance is subsequently evaluated. However, upon completion of the curing process, all samples infused with high-flow repair materials exhib-ited cracking from the center, as illustrated in Fig 15. This phenomenon occurs because the soil inherently possesses a low water content, whereas the high-flow repair materials contain a significantly higher water content. Consequently, the soil rapidly absorbs moisture from the newly infused repair materials, accelerating water loss within these materials and thereby inducing dry cracking. Through continuous testing, to avoid this phenomenon, the sample was pre-treated before refilling. Specifically, approximately 5 ml of water was dripped onto both sides of the crack to ensure adequate absorption by the surrounding soil. Subsequently, the repair material was added to the crack, which effectively reduced the absorp-tion of water from the grouting material by the soil on both sides of the crack and allowed the material's water content to evaporate normally. After pretreatment, the effect is illustrated in Fig 14b, and the repair material maintains good integrity.

**(2) Freezing and thawing test of jointed samples**

For the repaired samples, based on the fluctuations in the highest and lowest temperatures in Kaifeng, the temperature range for the freeze-thaw cycle is set to −10°C to 20°C. Each cycle lasts 24 hours, comprising 12 hours of freezing and 12 hours of thawing. A total of 30 freeze-thaw cycles was conducted to simulate the extreme weather conditions in winter in the Kaifeng area, thereby evaluating the durability of repaired cracks and the compatibility between the repair materials and the native soil.

Fig 15 illustrates the crack repair of soil samples following different numbers of freeze-thaw cycles. In the fig., (a), (b), (c), and (d) correspond to samples subjected to 5, 21, 27, and 30 freeze-thaw cycles, respectively. By examining these images, the development of cracks and the effectiveness of repair in soil samples under varying freeze-thaw cycles can be analyzed.

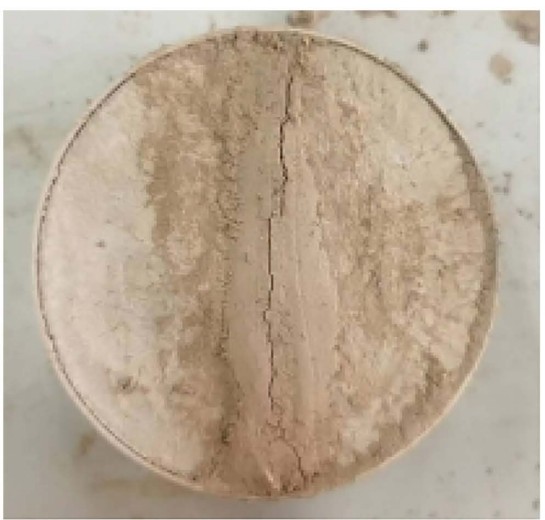 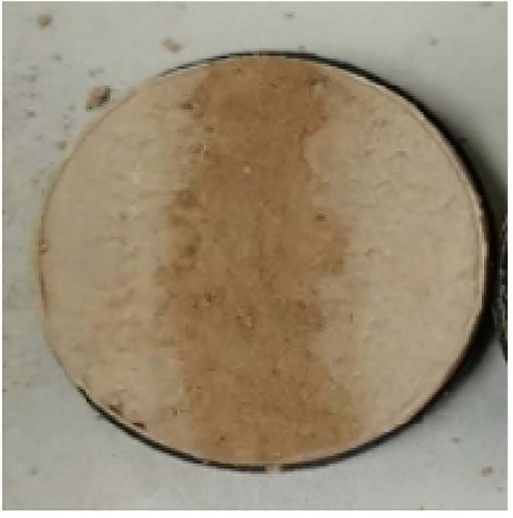

(a) Before treatment          (b) after processing

**Fig 14. Effect of different treatment methods on crack repair.**

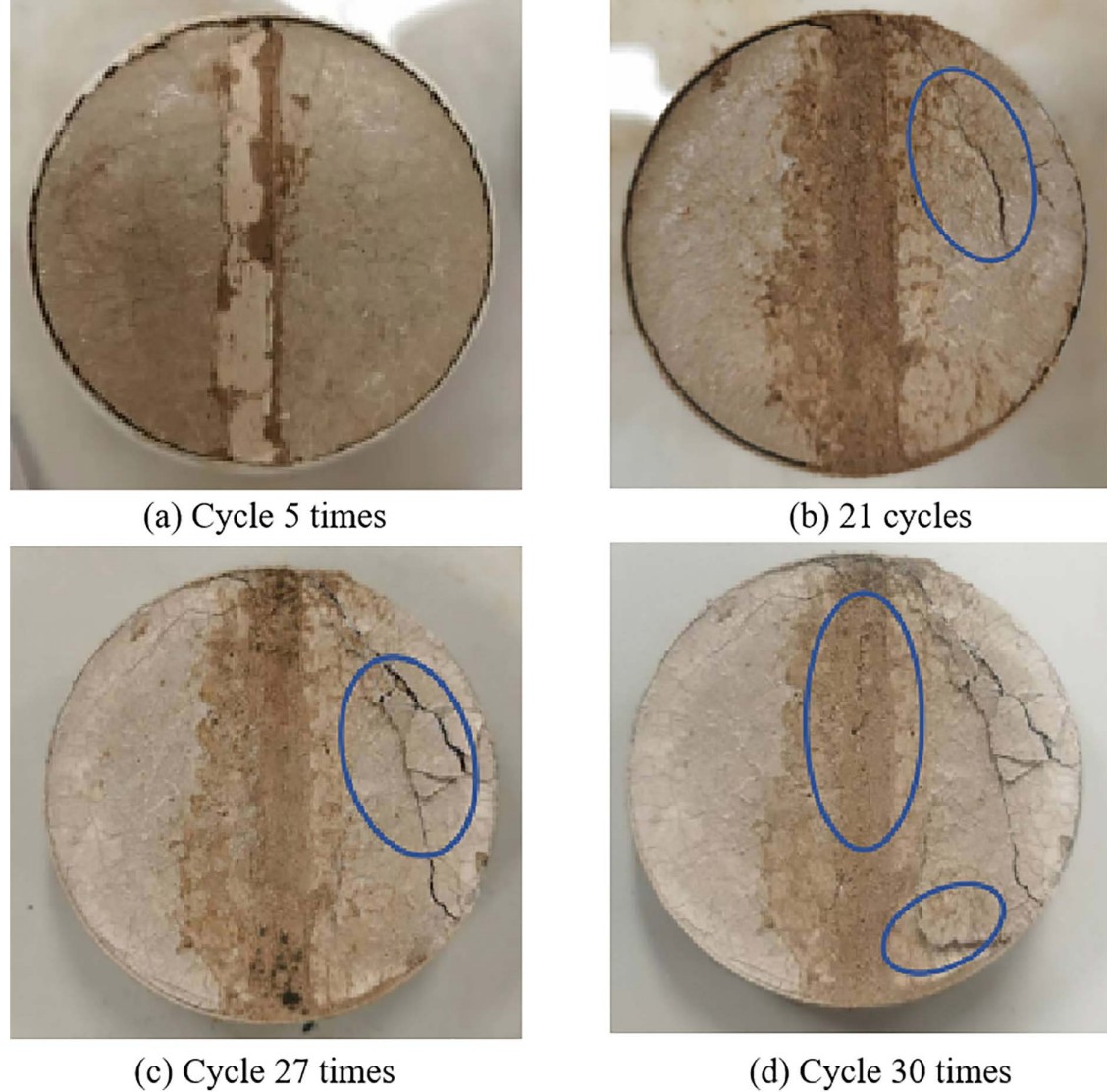

(a) Cycle 5 times  (b) 21 cycles

(c) Cycle 27 times  (d) Cycle 30 times

**Fig 15. Cracking after freeze-thaw cycle.**

Firstly, as shown in Fig 15a, when the number of freeze-thaw cycles is five, the repaired cracks remain intact overall. The water absorption of the repair material at the crack location is significantly lower than that of the surrounding soil. The surrounding soil is fully saturated with water, whereas the repair material exhibits minimal water absorption, with only slight saturation observed at the repair site. At the junction, no significant cracking is observed after the freeze-thaw cycle, indicating that the repair materials and the repair process are effective under these conditions.

With the increasing number of freeze-thaw cycles, as illustrated in Fig 15b, after 21 cycles, the repaired area maintains good integrity. However, the surrounding soil, due to its limited water resistance and durability, exhibits minor cracks and signs of damage following repeated freeze-thaw cycles. The remaining groups remain intact overall. It can be observed that at this stage, the anti-freeze-thaw cycle capability of the repair material exceeds that of the surrounding soil near the crack, indicating that the repair material possesses excellent durability.

Further increasing the number of freeze-thaw cycles to 27, as depicted in Fig 15c. After 27 cycles, no cracking is observed in the repaired sections of all groups. However, the cracking in the surrounding soil becomes more severe, and phenomena such as spalling and peeling are evident. This further highlights the necessity of timely repair for earthen sites affected by crack-related issues. These micro-cracks may result from the further accumulation and release of internal stress in soil samples, leading to additional damage around the cracks in the soil samples.

Finally, as shown in Fig 15d, when the number of freeze-thaw cycles reaches 30, tiny cracks begin to appear at the repair site and gradually deteriorate. However, compared with the cracks in the surrounding soil, these cracks are smaller, and the repaired area still exhibits superior mechanical properties and durability.

In summary, the crack repair materials exhibit excellent durability and compatibility with the soil. After 30 freeze-thaw cycles, they continue to maintain robust integrity, and no new cracks have emerged on either side of the repair site. As the number of freeze-thaw cycles increases, the crack propagation rate in soil samples accelerates, and both the number of cracks and their distribution range increase significantly. The research findings hold significant importance for understanding and predicting the mechanical behavior and durability of repair materials and repaired samples in freeze-thaw environments. Furthermore, they provide a scientific basis for the engineering application and protective measures of repairing earthen sites in such environments.

The properties of earthen site restoration materials are primarily influenced by the chemical reaction products and reaction kinetics of urease, urea, quicklime, and sodium methylsilicate. The mineralization reaction of urease with urea and quicklime involves the decomposition of urea to produce $CO_3^{2-}$ and $CaCO_3$ in conjunction with $Ca^{2+}$ from hydrated quicklime. This process constitutes a key area of investigation for many scholars studying urease mineralization as a means to enhance the mechanical properties of soil. In this study, calcium ions generated during the maturation of quicklime are employed as a calcium source. This approach not only eliminates the additional cost associated with using calcium chloride but also prevents environmental pollution caused by calcium ion leaching. Furthermore, this study leverages the principle that $CO_2$ produced during urease mineralization facilitates the reaction of sodium methylsilicate, thereby enhancing the carbonation reaction of sodium methylsilicate, improving its utilization efficiency, and strengthening the waterproofing performance of the repair materials. Thirdly, the presence of hydrated quicklime in the repair materials plays a critical role. The carbonation reaction between hydrated quicklime and atmospheric $CO_2$ serves as the foundation for enhancing the soil wall's properties. Additionally, the $CO_2$ generated from the urease-urea reaction further promotes the mineralization reaction of hydrated quicklime, thereby improving the mechanical properties near cracks and within the repair materials, increasing the compactness of the repair materials, enhancing the effectiveness and efficiency of crack repair, and reducing the likelihood of crack formation.

## 5. Discussion

This research centered on the repair needs of the crack pathologies in the earthen city wall of Kaifeng. It systematically evaluated the mechanical and durability properties of high-fluidity repair materials under various combinations of quicklime dosages and admixtures (sodium methylsilicate, styrene-acrylic emulsion, and waterborne polyurethane). The results indicated that the group with 7.5% quicklime and 7% sodium methyl silicate exhibited the optimal comprehensive performance. The mechanism lies in the fact that the urease-mediated mineralization reaction of urea and quicklime involves the decomposition of urea to generate $CO_3^{2-}$, which subsequently reacts with $Ca^{2+}$ in slaked quicklime to form $CaCO_3$. This process has been a key focus in the research on urease-induced mineralization by numerous scholars, to enhance the mechanical properties of soil. In this study, the calcium ions generated from the hydration of quicklime were employed as the calcium source. This approach not only circumvented the additional costs associated with the use of calcium chloride but also mitigated the environmental pollution caused by calcium ion leaching. Secondly, this study exploited the principle that the $CO_2$ generated during urease-mediated mineralization promotes the reaction of sodium methyl silicate. This effectively enhanced the carbonation reaction of sodium methyl silicate, thereby improving its utilization efficiency and further enhancing the waterproof performance of the repair material. Thirdly, the carbonation reaction between slaked

quicklime in the repair material and $CO_2$ in the air is crucial for the improvement of the earthen city wall. The $CO_2$ produced from the reaction of urease and urea also stimulates the mineralization reaction of slaked quicklime. This process not only strengthens the mechanical properties of the region surrounding the cracks and the repair material but also increases the compactness of the repair material. Consequently, it significantly enhances the effectiveness and efficiency of crack repair and reduces the likelihood of crack recurrence.

During the freeze-thaw cycle tests, the repair material exhibited greater durability compared to the surrounding soil mass. Notably, it retained relatively good integrity even after 30 freeze-thaw cycles. This finding indicates that the high-fluidity repair material demonstrates excellent adaptability under extreme climatic conditions. However, it should be emphasized that under prolonged freeze-thaw actions, the repaired area may still develop minor cracks. Consequently, in practical engineering applications, it is necessary to implement comprehensive protective measures tailored to the environmental conditions. For instance, a combination of surface protection and internal reinforcement techniques should be considered. However, the experimental results are predominantly derived from indoor simulated environments and thus do not comprehensively mirror the intricate environmental factors of actual earthen heritage sites. These factors include long-term wet-dry cycles, salt migration, and biological influences, among others. Consequently, future research endeavors should, based on field experiments and long-term monitoring, further validate the applicability and durability of the repair materials within real-world settings.

Moreover, there are several practical challenges associated with the on-site application of repair materials. Firstly, cost represents a notable hurdle. Certain admixtures, such as waterborne polyurethane and styrene-acrylic emulsion, come at a relatively high price. Consequently, large-scale utilization may be constrained by budgetary limitations. Secondly, compatibility is a crucial aspect. The disparities in composition, pore structure, and moisture content among the soils of different heritage sites, along with the varying environmental conditions where these sites are situated, can all exert an influence on the permeability and bonding properties of the repair materials. This, in turn, has an impact on the overall repair outcome. Inadequate handling may even give rise to new stress concentrations between the repair materials and the original soil matrix, potentially leading to secondary damage. Therefore, when it comes to promoting and implementing these repair materials, it is imperative to formulate personalized repair plans that take into account the specific environmental conditions of local heritage sites.

Generally speaking, this study not only furnishes a scientific foundation for the restoration of crack-related pathologies in the earthen city wall of Kaifeng but also holds significant reference value for the selection and optimization of repair materials in the conservation of other earthen heritage sites. Nevertheless, to effectuate an efficient transition from laboratory findings to on-site applications, it is essential to conduct further systematic validations in consideration of the intricate environmental conditions specific to each heritage site.

## 6. Conclusions

To develop repair materials for soil wall cracks with good fluidity, water resistance, no cracking, high strength, and a color similar to the soil wall, this study investigates the surface strength, water absorption, post-water absorption surface strength, and consistency of 27 groups of high-flow repair materials. These materials vary in quicklime content, sodium methylsilicate concentration, styrene-acrylic emulsion, and waterborne polyurethane solution. Additionally, the practical repair effectiveness of these high-flow materials was verified. The following conclusions were drawn:

(1) The mineralization of urease urea in combination with quicklime and sodium methylsilicate accelerates the chemical reaction between quicklime and sodium methylsilicate, thereby enhancing the mechanical and waterproofing properties of restoration materials. This process offers a novel approach for developing materials suitable for soil wall restoration.

(2) Sodium methylsilicate, styrene-acrylic emulsion, and waterborne polyurethane can enhance the consistency of fluidized repair materials, whereas quicklime serves to decrease material consistency. With a specific quicklime content,

the consistency of repair materials increases as the concentration of sodium methylsilicate and waterborne polyure-thane rises, yet decreases with an increase in the concentration of styrene-acrylic emulsion.

(3) All three solutions enhance the waterproof performance of high-flow repair materials. In terms of overall waterproof performance, waterborne polyurethane outperforms sodium methylsilicate, which in turn surpasses styrene-acrylic emulsion. The lowest water absorption rate, at 3.9%, was observed in the group with a 5% sodium methylsilicate concentration and a 7.5% quicklime content. As the quicklime content increases, the waterproof performance of each group declines to varying degrees; however, all groups still exhibit relatively high waterproof performance.

(4) Quicklime, styrene-acrylic emulsion, and waterborne polyurethane solution significantly enhance the strength of high-fluidity repair materials. Under the same concentration, the surface strength initially increases and subsequently decreases with the increasing quicklime content. When the quicklime content is held constant, the surface strength increases with the increasing concentration of styrene-acrylic emulsion and initially increases before subsequently decreasing with the increasing concentration of waterborne polyurethane.

(5) The overall properties of the 27 groups of high-flow restoration materials vary. When in use, considering the dif-fering environmental, climatic, and groundwater conditions across regions, the efficiency coefficient method can be employed to adjust the weighting of various properties, thereby identifying the optimal restoration materials and enhancing the crack restoration of immovable cultural relics in diverse regions. The total efficacy coefficient of the group containing 7.5% quicklime and 7% sodium methylsilicate is 96.25, indicating that it possesses excellent mechanical properties and waterproof performance.

In the joint filling test, water should be added to moisten the cracks, ensuring they are not in a dry state, thereby enhancing the bonding strength and restoration efficacy between the repair materials and the earthen site. Thirty crack-filling experiments demonstrate that the crack resistance of the filling materials is superior to that of the samples, and the filling materials exhibit enhanced resistance to freeze-thaw cycles. Currently, the research primarily focuses on the development of repair materials, with experimental studies conducted on indoor samples. In the future, it will be necessary to investigate the influence of various factors (such as temperature, humidity, chemical erosion, etc.) on the performance of crack repair materials based on the environmental conditions of earthen sites, thereby enabling a more comprehensive evaluation of material performance in real-world scenarios. A long-term durability test should also be conducted to assess the performance changes of crack repair materials during extended use, including the effects of aging, erosion, and other relevant factors.

## Author contributions

**Conceptualization:** Jianwei Yue.

**Data curation:** Tingting Yue, Wenhao Li, Xiang Zhu.

**Formal analysis:** Tingting Yue, Wenhao Li, Xiang Zhu.

**Methodology:** Jianwei Yue, Jingwen Yue.

**Project administration:** Xizhi Zhang.

**Supervision:** Xizhi Zhang, Jianwei Yue.

**Validation:** Jingwen Yue.

**Visualization:** Wenhao Li, Xiang Zhu.

**Writing – original draft:** Tingting Yue.

**Writing – review & editing:** Tingting Yue.

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
