## [Decision Letter · Decision Letter 0]

15 Apr 2025

PONE-D-25-02028
Research on repairing materials and techniques of cracks in Kaifeng earth wall
PLOS ONE

Dear Dr. 岳,

Thank you for submitting your manuscript to PLOS ONE. After careful consideration, we feel that it has merit but does not fully meet PLOS ONE’s publication criteria as it currently stands. Therefore, we invite you to submit a revised version of the manuscript that addresses the points raised during the review process.

We look forward to receiving your revised manuscript.

Kind regards,

Gobinath Ravindran

Academic Editor

PLOS ONE

**Journal Requirements:**

1. When submitting your revision, we need you to address these additional requirements.
 
Please ensure that your manuscript meets PLOS ONE's style requirements, including those for file naming. The PLOS ONE style templates can be found at 
https://journals.plos.org/plosone/s/file?id=wjVg/PLOSOne_formatting_sample_main_body.pdf and 
https://journals.plos.org/plosone/s/file?id=ba62/PLOSOne_formatting_sample_title_authors_affiliations.pdf
 
2. We suggest you thoroughly copyedit your manuscript for language usage, spelling, and grammar. If you do not know anyone who can help you do this, you may wish to consider employing a professional scientific editing service. 
 
The American Journal Experts (AJE) (https://www.aje.com/) is one such service that has extensive experience helping authors meet PLOS guidelines and can provide language editing, translation, manuscript formatting, and figure formatting to ensure your manuscript meets our submission guidelines. Please note that having the manuscript copyedited by AJE or any other editing services does not guarantee selection for peer review or acceptance for publication. 
 
Upon resubmission, please provide the following:
 
The name of the colleague or the details of the professional service that edited your manuscript
 
A copy of your manuscript showing your changes by either highlighting them or using track changes (uploaded as a *supporting information* file)
 
A clean copy of the edited manuscript (uploaded as the new *manuscript* file)
 
3. In your Methods section, please provide additional information regarding the permits you obtained for the work. Please ensure you have included the full name of the authority that approved the field site access and, if no permits were required, a brief statement explaining why.
 
4. We note that your Data Availability Statement is currently as follows: All relevant data are within the manuscript and its Supporting Information files.
 
Please confirm at this time whether or not your submission contains all raw data required to replicate the results of your study. Authors must share the “minimal data set” for their submission. PLOS defines the minimal data set to consist of the data required to replicate all study findings reported in the article, as well as related metadata and methods (https://journals.plos.org/plosone/s/data-availability#loc-minimal-data-set-definition).
 
For example, authors should submit the following data:
 
- The values behind the means, standard deviations and other measures reported;
- The values used to build graphs;
- The points extracted from images for analysis.
 
Authors do not need to submit their entire data set if only a portion of the data was used in the reported study.
 
If your submission does not contain these data, please either upload them as Supporting Information files or deposit them to a stable, public repository and provide us with the relevant URLs, DOIs, or accession numbers. For a list of recommended repositories, please see https://journals.plos.org/plosone/s/recommended-repositories.
 
If there are ethical or legal restrictions on sharing a de-identified data set, please explain them in detail (e.g., data contain potentially sensitive information, data are owned by a third-party organization, etc.) and who has imposed them (e.g., an ethics committee). Please also provide contact information for a data access committee, ethics committee, or other institutional body to which data requests may be sent. If data are owned by a third party, please indicate how others may request data access.
 
5. Please amend the manuscript submission data (via Edit Submission) to include authors Dr. Wenhao Li, Jingwen Yue and Xiang Zhu.
 
6. Please upload a new copy of Figure 3 as the detail is not clear. Please follow the link for more information: https://blogs.plos.org/plos/2019/06/looking-good-tips-for-creating-your-plos-figures-graphics/" https://blogs.plos.org/plos/2019/06/looking-good-tips-for-creating-your-plos-figures-graphics/"

Reviewers' comments:

Reviewer's Responses to Questions

**Comments to the Author**

1. Is the manuscript technically sound, and do the data support the conclusions?

Reviewer #1: Yes

Reviewer #2: Partly

2. Has the statistical analysis been performed appropriately and rigorously? 

Reviewer #1: No

Reviewer #2: No

3. Have the authors made all data underlying the findings in their manuscript fully available?

Reviewer #1: Yes

Reviewer #2: Yes

4. Is the manuscript presented in an intelligible fashion and written in standard English?

Reviewer #1: Yes

Reviewer #2: No

5. Review Comments to the Author

**Reviewer #1:** The manuscript presents a valuable study on the repair of cracks in Kaifeng's earth walls using various materials, but there are several areas that require significant improvement before publication. One major concern is the lack of a clear justification for the research’s novelty. While the introduction provides a detailed background on the deterioration mechanisms of earth walls (Page 2, Lines 10–38), it does not sufficiently differentiate this study from previous works. Many references (e.g., Cui et al. [3,4], Zhang et al. [8,9]) discuss similar approaches using strengthening agents, but the authors fail to explicitly state how their methodology or findings contribute to advancing the field. The manuscript would benefit from a stronger argument on what makes this study unique, such as the combination of repair materials or the specific application of the efficacy coefficient method.

The methodology section lacks crucial details that would allow replication. For example, while the authors describe the preparation of soil samples and repair materials (Page 6, Lines 12–40), they do not provide sufficient information about the curing conditions, environmental parameters, or the exact procedure for mixing and applying the materials. Additionally, while the freeze-thaw cycle test is mentioned (Page 12, Lines 4–18), the manuscript does not specify the temperature range, cycle duration, or the reasoning behind the number of cycles chosen. Given that freeze-thaw durability is a key performance indicator for long-term crack repair, these missing details significantly weaken the reliability of the results. The authors should provide explicit temperature settings and cycle descriptions, referencing standard testing protocols where applicable.

The discussion of results lacks depth in terms of explaining the chemical and physical interactions between repair materials and the soil. For instance, while the manuscript explains that sodium methylsilicate forms a hydrophobic film (Page 10, Lines 30–38), it does not explore whether this film affects the soil’s permeability or long-term stability. Additionally, the interaction between quicklime, urease, and urea in forming calcium carbonate (Page 7, Lines 20–38) is described, but the implications of this reaction on the material’s porosity and potential shrinkage are not addressed. Furthermore, no information is provided on possible secondary effects, such as efflorescence, which could impact the aesthetic and mechanical properties of the repaired cracks. A more detailed chemical discussion is needed to strengthen the conclusions.

There are several inconsistencies and grammatical errors throughout the manuscript. For instance, "methicosilicate" appears multiple times (Page 3, Line 20; Page 8, Line 5) but is likely meant to be "methylsilicate." Similarly, "waterborne polyurethane" and "water-based polyurethane" are used interchangeably (Page 3, Line 15; Page 9, Line 30), which could confuse readers. Terminology should be consistent throughout the paper. The abstract (Page 1, Lines 10–25) also contains awkward phrasing, such as "the repair material does a great job fixing the simulated crack," which should be revised for a more formal academic tone. Furthermore, certain sentences are overly informal, such as "this stuff is pretty strong" (Page 2, Line 14), which should be rewritten in a more precise scientific manner.

Figures and tables require improvements in clarity. Some figures, such as the particle gradation diagram (Page 7, Figure 1), lack clear axis labels, making it difficult to interpret trends. Tables summarizing experimental data should include statistical comparisons to highlight significant differences between groups. Additionally, some references in the bibliography appear incomplete or improperly formatted. For example, certain citations (e.g., Page 19, References [12,13]) lack full journal details and DOI links. Ensuring correct formatting and consistency across citations is essential for maintaining academic credibility.

**Reviewer #2: **I have gone through the manuscript and the topic is good. I have some recommendations that would help to improve the quality of the manuscript.

1. A graphical abstract should be included in the manuscript to make it more appealing.

2. The abstract is too vague and does not summarize key findings properly.

3. The manuscript addresses an important topic, but its novelty needs to be clearly stated.

4. There is a lack of clear objectives; these should be explicitly stated.

5. There are a lot of headings in the manuscript, which does not make sense. These should be removed, and flow and coherence should be maintained.

6. The results are not well-organized and should be presented more systematically.

7. The authors should provide a stronger closing statement.

8. The aesthetics could be improved in all of the images and graphs.

9. All the headings should be revised and made more TO THE POINT and brief.

10. All the figures need revision. All the figures have the potential that a lot of information may be included in them. They should be made more informatically appealing with more attention to detail.

11. The design of experiments (DOE) should be more clearly explained.

12. The conclusion section should be more robust and clearly state the findings of the results obtained. Future scope and work should be included.

13. The Results and Discussion section should be added and populated as necessary.

14. The limitations and the future potentials of the study are not sufficiently addressed or acknowledged.

6. PLOS authors have the option to publish the peer review history of their article (what does this mean?). If published, this will include your full peer review and any attached files.

Reviewer #1: **Yes: **Muhammad Usama Zaheer

Reviewer #2: No

---

## [Author Response · Author response to Decision Letter 1]

21 Jun 2025

We really appreciate the time the reviewers have spent on our manuscript. The constructive and helpful suggestions are valuable for us to improve the quality of the manuscript. We have revised the manuscript carefully according to all the comments. All the changes are marked with highlight tool in the revised manuscript.

---

## [Decision Letter · Decision Letter 1]

9 Sep 2025

PONE-D-25-02028R1
Study on Repair Materials and Technologies for Addressing Crack-Related Damage in the Earthen City Wall of Kaifeng
PLOS ONE

Dear Dr. 岳,

Thank you for submitting your manuscript to PLOS ONE. After careful consideration, we feel that it has merit but does not fully meet PLOS ONE’s publication criteria as it currently stands. Therefore, we invite you to submit a revised version of the manuscript that addresses the points raised during the review process.
 
Your manuscript has been assessed by two reviewers, who both raise points and concerns to address. Please give these points careful consideration when preparing your revised manuscript and point-by-point response document.

We look forward to receiving your revised manuscript.

Kind regards,

Dr Jason Morgan

Staff Editor

PLOS ONE

Journal Requirements:

Reviewers' comments:

Reviewer's Responses to Questions

**Comments to the Author**

1. If the authors have adequately addressed your comments raised in a previous round of review and you feel that this manuscript is now acceptable for publication, you may indicate that here to bypass the “Comments to the Author” section, enter your conflict of interest statement in the “Confidential to Editor” section, and submit your "Accept" recommendation.

Reviewer #1: (No Response)

Reviewer #3: All comments have been addressed

2. Is the manuscript technically sound, and do the data support the conclusions?

Reviewer #1: Partly

Reviewer #3: No

3. Has the statistical analysis been performed appropriately and rigorously? 

Reviewer #1: No

Reviewer #3: No

4. Have the authors made all data underlying the findings in their manuscript fully available?

Reviewer #1: Yes

Reviewer #3: No

5. Is the manuscript presented in an intelligible fashion and written in standard English?

Reviewer #1: No

Reviewer #3: No

6. Review Comments to the Author

Reviewer #1: The manuscript presents an experimental study on repair materials and technologies for addressing crack-related damage in the Kaifeng earthen city wall. The topic is relevant and of practical importance to cultural heritage conservation. The experimental program is broad, testing 27 combinations of materials under multiple performance indicators, and the findings have practical significance for field applications. However, several issues should be addressed before the manuscript can be considered suitable for publication.

Language and Readability

The manuscript requires thorough English editing to correct grammatical errors, improve clarity, and reduce repetition. Several “Error! Reference source not found.” placeholders remain in the text and must be fixed. Improving the overall readability will significantly strengthen the manuscript’s presentation.

Novelty and Positioning

While the experimental program is systematic, the novelty compared to existing studies is not strongly emphasized. Many of the repair materials (quicklime, sodium methylsilicate, polyurethane, styrene-acrylic emulsion) and techniques (EICP/MICP) have already been studied in the context of soil stabilization and conservation. The authors should clearly highlight:

What specific research gap this study addresses.

How the work differs from and advances beyond prior studies.

Whether the innovation lies in the optimization of material ratios, the multi-indicator evaluation method, or the integration of hydraulic and mechanical performance metrics.

A deeper comparison with recent international literature is recommended.

Figures and Tables

Several figures and tables are of low quality, unclear, or missing proper references. All figures should be redrawn at higher resolution with consistent labeling. Broken cross-references must be corrected. Ensure that captions are detailed enough to allow standalone interpretation.

Statistical Analysis and Rigor

The results are largely presented as single values without error bars, standard deviations, or information on replicates. For a journal such as PLOS ONE, statistical robustness is essential. Please clarify the number of replicates for each test and include appropriate measures of variability. If possible, apply statistical tests (e.g., ANOVA) to support claims of significant differences.

Discussion Section

The discussion is mainly descriptive. It should go beyond reporting test outcomes and provide deeper insights into why certain combinations performed better. For example, what mechanisms explain the superior performance of 7.5% quicklime + 7% sodium methylsilicate? How do these results align with or contradict existing studies? The authors should also discuss limitations (e.g., scale effects, laboratory vs. field conditions) and the potential implications for conservation practices.

Reviewer #3: Recommendations for Improvement

1. Improve Clarity and Structure:

o Revise the introduction to clearly state the research problem, objectives, and the specific gap this study addresses compared to existing literature.

o Organize the manuscript into well-defined sections: Introduction, Literature Review, Methodology, Results, Discussion, and Conclusion. Ensure each section flows logically and avoids redundancy.

o Include a graphical abstract, as suggested by Reviewer 2, to provide a visual summary of the study's key findings and methodology.

2. Enhance Methodological Details:

o Provide a detailed explanation of the DOE, including the rationale for selecting specific concentrations of quicklime and waterproofing agents. Justify the choice of 27 groups and the criteria for material selection.

o Clarify the efficacy coefficient method by providing the mathematical formulation, assumptions, and how weighting coefficients were determined.

o Include a section on quality control measures during sample preparation and testing to ensure reproducibility.

3. Improve Data Presentation:

o Ensure all tables and figures are complete, with clear captions, axis labels, and statistical comparisons (e.g., ANOVA or t-tests) to validate significant differences between groups.

o Revise incomplete or poorly formatted tables (e.g., Table 5) and figures (e.g., particle gradation diagram). Provide a summary table comparing the performance of all 27 groups across the tested indicators.

o Include error bars or confidence intervals in graphical representations to indicate data variability.

4. Address Environmental Factors:

o Expand the discussion to include the coupled effects of temperature fluctuations, humidity, and groundwater dynamics, as suggested already. Consider conducting additional experiments or referencing relevant studies to address these factors.

o Provide a more detailed explanation of how the freeze-thaw cycle test simulates real-world conditions in Kaifeng.

5. Correct Formatting and Language:

o Thoroughly proofread the manuscript to correct grammatical errors, awkward phrasing, and OCR-related issues (e.g., "5.5.5.5" or "$\mathrm{J} \mathrm{J} \mathrm{J}$").

o Ensure consistent formatting of chemical names (e.g., "styrene-acrylic emulsion" instead of "styrene-acylle") and equations. Verify that all mathematical expressions are correctly rendered.

o Complete and standardize the reference list, ensuring all citations include necessary details (e.g., DOIs where available).

Specific Comments

• Abstract: If a graphical abstract has been added , ensure it is clear and visually appealing. The written abstract should concisely summarize the study's objectives, methods, key findings (e.g., 3.9% water absorption for the 5% sodium methylsilicate + 7.5% quicklime group), and implications.

• Introduction: Strengthen the literature review by citing more recent studies to contextualize the research gap.

• Methodology : Provide a flowchart or diagram to illustrate the experimental workflow, including sample preparation, testing procedures, and data analysis.

• Results : The claim of a 3.9% water absorption rate is significant but needs statistical validation. Discuss why sodium methylsilicate outperforms other waterproofing agents in greater detail.

• Discussion : Elaborate on the practical challenges of applying these repair materials in situ, such as cost, scalability, and compatibility with existing structures.

• References: Ensure all references (e.g., [1], [25], [40]) are complete and follow the journal's formatting guidelines.

7. PLOS authors have the option to publish the peer review history of their article (what does this mean?). If published, this will include your full peer review and any attached files.

Reviewer #1: No

Reviewer #3: **Yes: **amirbahram arabahmadi

---

## [Author Response · Author response to Decision Letter 2]

25 Sep 2025

We sincerely appreciate your meticulous review of our manuscript and the invaluable feedback provided. We hold your comments and suggestions in the highest regard and have accordingly made comprehensive revisions to the manuscript.

---

## [Editor Report · Decision Letter 2]

2 Oct 2025

Study on Repair Materials and Technologies for Addressing Crack-Related Damage in the Earthen City Wall of Kaifeng

PONE-D-25-02028R2

Dear Dr. 岳,

We’re pleased to inform you that your manuscript has been judged scientifically suitable for publication and will be formally accepted for publication once it meets all outstanding technical requirements.

Kind regards,

Hailing Ma

Academic Editor

PLOS ONE
---

## [Editor Report · Acceptance letter]

PONE-D-25-02028R2

PLOS ONE

Dear Dr. Yue,

I'm pleased to inform you that your manuscript has been deemed suitable for publication in PLOS ONE. Congratulations! Your manuscript is now being handed over to our production team.

Kind regards,

on behalf of

Dr. Hailing Ma

Academic Editor

PLOS ONE